# Spindle integrity is regulated by a phospho-dependent interaction between the Ndc80 and Dam1 kinetochore complexes

**Christian R. Nelson**[1], **Darren R. Mallett**[1,2], **Sue Biggins**[1]*

**1** Howard Hughes Medical Institute, Division of Basic Sciences, Fred Hutchinson Cancer Center, Seattle, Washington, United States of America, **2** Molecular and Cellular Biology Program, University of Washington, Seattle, Washington, United States of America

* sbiggins@fredhutch.org

## Abstract

Faithful chromosome segregation depends upon kinetochores, large protein complexes that anchor chromosomes to dynamic microtubules, allowing for their movement at anaphase. Critical microtubule-coupling components of the budding yeast kinetochore, the Dam1 (Dam1c) and Ndc80 (Ndc80c) complexes, work cooperatively to ensure that kinetochores track with the plus-ends of microtubules. Additionally, the Dam1 complex plays a distinct role in ensuring the integrity of the mitotic spindle. However, the events required to orchestrate these diverse functions of Dam1c remain unclear. To identify regulatory events on kinetochores, we performed phosphoproteomics on purified kinetochore proteins and identified many previously unknown phosphorylation events. We demonstrate that Ndc80 is phosphorylated at Thr-248 and Thr-252 to promote the interaction between Ndc80 and the Dam1c. The phosphorylation of T248 is cell cycle regulated and depends on Mps1. Ndc80 phosphorylation at T248 and T252 does not appear to regulate kinetochore function and instead contributes to Dam1c localization to the anaphase spindle. A *ndc80* phospho-deficient mutant exhibited a genetic interaction and altered spindle morphology when combined with *dam1* mutant alleles. Taken together, we propose that Mps1-dependent phosphorylation of Ndc80 at T248 and T252 is removed at anaphase to allow Dam1c to help organize and stabilize the spindle.

## Author summary

Cells must inherit the right number of chromosomes every time they divide. Defects in this process are the most common chromosomal abnormality in cancer cells and contribute to birth defects and other diseases. We therefore study the process of chromosome partitioning to daughter cells when they divide to understand the basis for many human diseases. The machine that moves chromosomes is called the kinetochore. It is made up of many proteins that assemble onto chromosomes and attach to filaments called microtubules that form the spindle. Here, we identify a previously unknown regulatory event that ensures that two kinetochore complexes interact at the appropriate time in the cell cycle. The interaction is decreased when chromosomes move to opposite cells to allow one of the complexes to bind to microtubules to facilitate spindle function.

**Data availability statement:** Mass spectrometry data generated in this study is available through Mass Spectrometry Interactive Virtual Environment (MassIVE, University of California San Diego) located here (https://massive.ucsd.edu/ProteoSAFe/dataset.jsp?task=ed-f1adb3ff8d440f8aa483f86e1034a4). The code generated for this study is available here at Github: https://github.com/darrenrmallett/ms-phos-analysis-Nelson-et-al

**Funding:** This research was supported by the NIH P30 CA015704 award to the Proteomics and Metabolomics Shared Resource, RRID:SCR_022618, of the Fred Hutch/University of Washington/Seattle Children's Cancer Consortium. This work was supported by an NIH (R35 GM149357) grant and the Howard Hughes Medical Institute investigator award to S.B. The funders had no role in study design, data collection and analysis, decision to publish, or preparation of the manuscript.

## Introduction

Accurate chromosome segregation relies on kinetochores, megadalton protein structures that attach chromosomes to dynamic microtubules to orchestrate cell division. Defects in chromosome segregation generate aneuploidy, a hallmark of tumor cells and other diseases [1–3]. Kinetochores are made up of distinct regions: centromeric chromatin that is defined by the centromeric histone H3 variant [4–7], the outer kinetochore that directly interacts with force-generating microtubules [8–10], and the inner kinetochore (CCAN) that bridges the two [11,12]. Proper chromosome segregation requires sister kinetochores to bi-orient and attach to microtubules from opposite spindle poles to ensure that each daughter cell receives an equivalent set of chromosomes. However, sister kinetochores initially often make incorrect attachments that are sensed by a molecular surveillance system called the spindle assembly checkpoint that halts the cell cycle until the errors are corrected [13–15]. The outer kinetochore is both the target of error correction mechanisms and the scaffold for the spindle assembly checkpoint.

Phosphorylation serves key regulatory roles in a diverse set of kinetochore functions, such as the spindle assembly checkpoint, error correction, and kinetochore assembly [16]. Two conserved kinases critical to kinetochore phosphoregulation are Mps1 and Aurora B (Ipl1 in budding yeast) [17]. Both kinases mediate various aspects of kinetochore assembly. For example, Aurora B phosphorylation of the Dsn1 protein relieves autoinhibition of a kinetochore binding site in the conserved Mtw1 complex [18–20]. The Mps1 kinase initiates the spindle assembly checkpoint via phosphorylation of the outer kinetochore protein Spc105 [21–23]. Mps1 and Aurora B are also both essential for error correction and phosphorylate the major microtubule binding complex Ndc80 (Ndc80c) [24–29]. In yeast, mutations in the Ndc80 phosphorylation sites are not lethal, indicating that other kinase targets exist. For Ipl1, the other major error correction target is the microtubule-interacting complex, Dam1 (Dam1c) [8,30–33]. However, additional Mps1 error correction targets have not yet been identified.

In yeast, the Ndc80 and Dam1 subcomplexes directly interact and form the major yeast kinetochore-microtubule interface [34–36]. While structural studies have provided insight into the Dam1c-Ndc80c interaction, some regions are missing in the structures, so all the precise interactions are not yet known [37,38]. The four protein Ndc80c forms an elongated dumbbell-like tetrameric structure, with the globular calponin-homology domain of Ndc80 interacting with the microtubule [10,39,40]. The disordered N-terminal tail of Ndc80 also associates with the microtubule and phosphorylation of this domain by Aurora B and Mps1 weakens the Ndc80-microtubule interaction to mediate error correction [9,28,41]. Dam1c is a 10-member complex that can form oligomers around the microtubule to help kinetochores bear chromosomal-load [42–44]. Ndc80 and Dam1 complexes work cooperatively: for example, Dam1c promotes Ndc80c affinity for dynamic microtubule tips and the Dam1c can compensate for the loss of the Ndc80 N-terminal tail in microtubule binding [34,35,45,46]. Moreover, the co-operativity of these two complexes is likely highly regulated—they must tightly bind dynamic microtubule tips, but not so tightly that improper attachments can't be corrected. Aurora B phosphorylation plays a role in this dynamicity [34,37,47], but it is unclear whether additional phosphorylation events are involved. Dam1c localizes to spindles in addition to kinetochores and has an independent function in proper mitotic spindle formation because mutants have defective spindles [48–50]. However, the precise function of Dam1c on the mitotic spindle and the regulatory events that ensure the proper association of Dam1c with the spindle remain largely unknown.

Here, we identified previously unknown budding yeast kinetochore phosphorylation with a focus on Mps1 sites since it is essential for error correction but its target(s) for this function remain elusive [27,29,51]. Our analyses identified Ndc80 phosphorylation at Thr-248 and

Thr-252 that are likely Mps1 regulated. Ndc80 phosphorylation promotes the Ndc80c-Dam1c interaction and dephosphorylation appears to help Dam1c localize to the mitotic spindle. Consistent with this, mutation of the *ndc80* phospho-sites exhibited genetic interactions with *dam1* complex alleles leading to spindle defects and chromosome mis-segregation. Together, our work identified a previously unknown regulatory event that contributes to the interaction of the major yeast microtubule binding complexes and suggests that characterizing the other phospho-sites we identified will further our understanding of the functions of kinetochore phosphorylation.

## Results

### Identification of phosphorylation on kinetochore proteins

We set out to identify native phosphorylation sites on kinetochore proteins that might provide insight into Mps1 kinase regulatory events. To this end, we used a previously established one-step Flag purification from strains containing 3Flag epitope tags on endogenous kinetochore components (Fig 1A) [52,53]. We isolated native kinetochore particles from a strain expressing Dsn1-His-Flag, the Ndc80 complex from a strain expressing Ndc80-Flag, and the Dam1 complex from a strain expressing Dad1-Flag. The purifications were performed from asynchronously growing wildtype (WT) cells as well as from cells overexpressing Mps1 (*pGAL-MPS1*), which arrests cells in metaphase, and should amplify Mps1-mediated phosphorylation. We verified the purifications worked via silver stain PAGE analysis and detected clear phospho-shifts for some kinetochore proteins when Mps1 was overexpressed (S1 Fig). For example, the Mps1 substrates Dam1 and Ask1 [54–57] exhibited higher molecular forms (Dad1–3FLAG purification, red *, S1C Fig) when Mps1 was overexpressed.

To identify the co-purifying proteins and associated phosphorylation events, the samples were submitted for mass spectrometry (MS) analysis (Fig 1A). MS confirmed that native kinetochore proteins (for Dsn1-His-Flag) and outer kinetochore complexes (for Ndc80 and Dad1) were specifically purified. Surprisingly, many additional proteins co-purified with Dsn1 and Ndc80 when Mps1 was overexpressed (S1A and S1B Fig). MS analysis showed that the most abundant proteins unique to Mps1 overexpression purifications of Dsn1 and Ndc80 are components of the actin cytoskeleton (S1E Fig). Of the top thirteen most enriched MS hits in purifications from *pGAL-MPS1* cells, eleven are related to actin biology (S1E Fig, denoted with *), but it is unclear whether these interactions are physiological. Mps1 overexpression did not cause a similar effect in the Dad1 purification (S1C Fig).

We analyzed the MS data for phosphorylation events on kinetochore proteins in both the WT and *pGAL-MPS1* purifications and identified 189 phosphorylation sites in 26 kinetochore proteins (S1 Table and Fig 1B). We included any phosphorylation event that was detected with ≥2 peptide-spectrum matches (PSMs) and had a confidence score of ≥75%. We then searched each phosphorylation site and its surrounding sequence for known kinase consensus motifs to assign a putative kinase for each site (Fig 1A and S2 Table). Thirty-three phosphorylation sites matched the Cdk1 the consensus motif [S/T-P] (S1 Table and Fig 1C). Consistent with previous data, we detected phosphorylation on the CCAN member, Ame1, at S41 and S45, Cdk1 sites known to regulate its stability [58]. Our dataset also included 31 putative or known Ipl1 phospho-sites matching the consensus motif [K/R-X-S/T-Y] (where Y is any amino acid other than proline) (S1 Table and Fig 1C)[32]. For example, we detected Dam1 phosphorylation at S20, an Ipl1 site that regulates Dam1 complex oligomerization [32,36,59]. To identify putative Mps1 sites, we used the consensus motif [D/E/N/C-X-S/T-Y] (where Y is any amino acid besides glutamate and proline) and detected 65 potential Mps1 phospho-sites (S1 Table and Fig 1C) [60]. We confirmed phosphorylation on Spc105 MELT motif residue

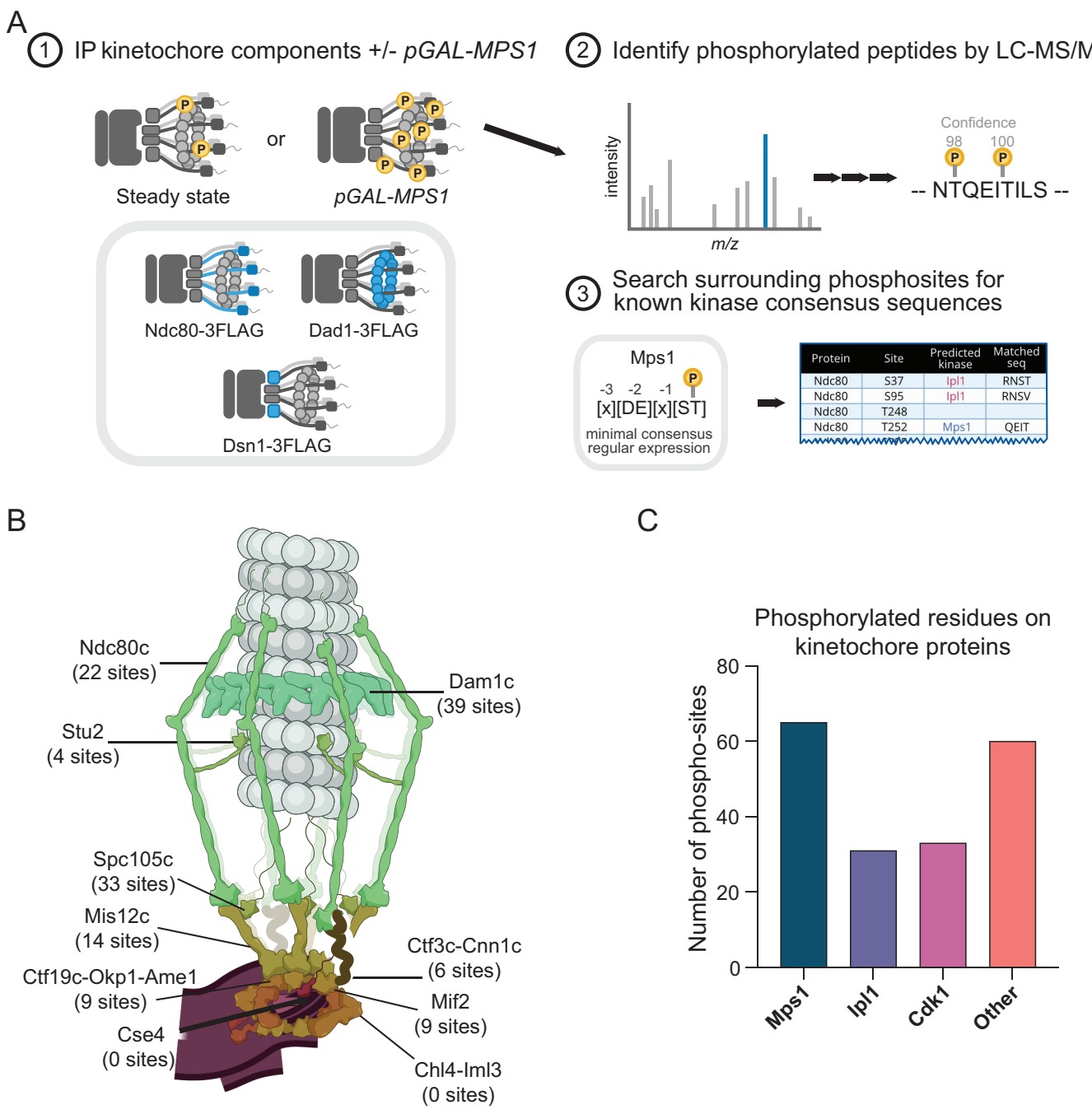

**Fig 1. Purifications of budding yeast kinetochore complexes reveal extensive phosphorylation.** (A) Schematic of method used to identify kinetochore phosphorylation. One-step purifications of Dsn1-Flag, Dad1-Flag, and Ndc80-Flag were performed in both WT and *pGAL-MPS1* genetic backgrounds. Puri-fied components were analyzed via mass spectrometry to identify phosphorylated residues. Computational motif analysis was performed to predict the most likely kinase responsible for phosphorylating each site. (B) Model of the budding yeast kinetochore with the number of phosphorylated sites detected in each subcomplex noted below. (C) Bar graph showing the number of phosphorylated residues on kinetochore proteins that match the consensus motif of Mps1, Aurora B (Ipl1) or Cdk1. "Other" represents phosphorylated residues on kinetochore proteins that do not fit any of the aforementioned consensus motifs.

T172 and identified many other sites in Spc105 matching the Mps1 consensus motif with -2 [S/T] phospho-priming (T59, S66, T83, T107, T111, S144, S250, S258, T355, T675). To further gauge which sites might be Mps1-dependent, we purified kinetochore particles from WT and

*mps1–1* cells to identify phosphorylation events that were lost on *mps1–1* kinetochore particles (Table 1 and S1D Fig). Although the absence of phosphorylation in this mass spec dataset is not conclusive that the site is Mps1-dependent, it may help to prioritize potential Mps1 sites. Consistent with this, phosphorylation of the Spc105 MELT motif, T172, was lost in the *mps1–1* mutant (Table 1). Taken together, these data identify many budding yeast kinetochore phosphorylation events and provide insight into the putative kinases.

## Mps1 phosphorylates Ndc80 at T248

We were particularly interested in Mps1-dependent phosphorylation of outer kinetochore complexes, given their potential role in microtubule-attachment and error correction. We previously found that Mps1 regulates Ndc80 via phosphorylation of its N-terminus at T74 to control kinetochore-microtubule attachment stability [28]. Two additional sites in Ndc80 that sit just C-terminal to the calponin-homology (CH) domain, T248 and T252, were phosphorylated in most of our datasets and were absent in the *mps1–1* dataset (Fig 2A and Tables 1 and S2). Given their proximity to the CH domain, these sites could be involved in regulating kinetochore-microtubule attachments. Furthermore, they are nearby a region previously shown to interact with Dam1 and could act as regulatory sites for one or more of these binding events [35] (Fig 2A). Ndc80-T252 follows the published Mps1 consensus motif [X-D/E-X-S/T] and we propose that T248 may also be regulated by Mps1, given that the Mps1 consensus motif is not rigid, and that Mps1 prefers to phosphorylate threonines preceded by [E/D/N/Q] at -2 [60]. These two sites are also in the same region of yeast Ndc80 as Human Hec1 S202, which fits the Mps1 consensus motif (Fig 2B). Hec1 Ser-202 was previously shown to be phosphorylated in multiple *in vivo* mass spectrometry studies, suggesting that the phosphorylation may be conserved [61,62]. Thus, we chose to further investigate Ndc80 T248 and T252 for a possible role in chromosome segregation.

To study these phospho-sites, we generated a phospho-antibody to Ndc80-T248 and verified that the antibody was specific. First, we tested whether the antibody recognizes Ndc80 when the phosphorylation site T248 is mutated to alanine. We mutated T252 in addition to T248 to facilitate phenotypic studies described below. We then purified Ndc80c (Spc24-His-Flag) from WT and *ndc80^{T248A,T252A}* cells and performed immunoblotting with the phospho-antibody. There was a strong Ndc80-T248 phospho-signal that was largely absent

**Table 1. Putative kinetochore Mps1 phosphorylation events.** Sites listed were detected in a purification of wildtype kinetochore particles (Dsn1-His-Flag) and fit the Mps1 consensus motif. The fraction of phosphorylated peptides relative to total peptides detected for a given site is shown for WT and *mps1-1* purifications. Bold phospho-sites showed lost or reduced phosphorylation in a purification from *mps1-1* cells.

| Subcomplex | Protein | Residue | WT phospho-peptides | *mps1–1* phospho-peptides |
|---|---|---|---|---|
| Ndc80 complex | Ndc80 | **T38** | 2/17 | 2/38 |
| | | **S97** | 2/4 | 0/3 |
| | | **T252** | 2/11 | 0/18 |
| Dam1 complex | Duo1 | **T30** | 2/6 | 0/4 |
| Spc105 complex | Spc105 | **T107** | 2/6 | 0/4 |
| | | T111 | 0/6 | 2/4 |
| | | **S250** | 2/4 | 0/4 |
| | | T355 | 0/7 | 7/16 |
| CENP-T | Cnn1 | S105 | 2/7 | no PSMs |
| CCAN | Mif2 | **S162** | 8/21 | 0/2 |
| | | S325 | 4/13 | no PSMs |
| Regulatory | Ubr2 | S717 | 2/4 | No PSMs |

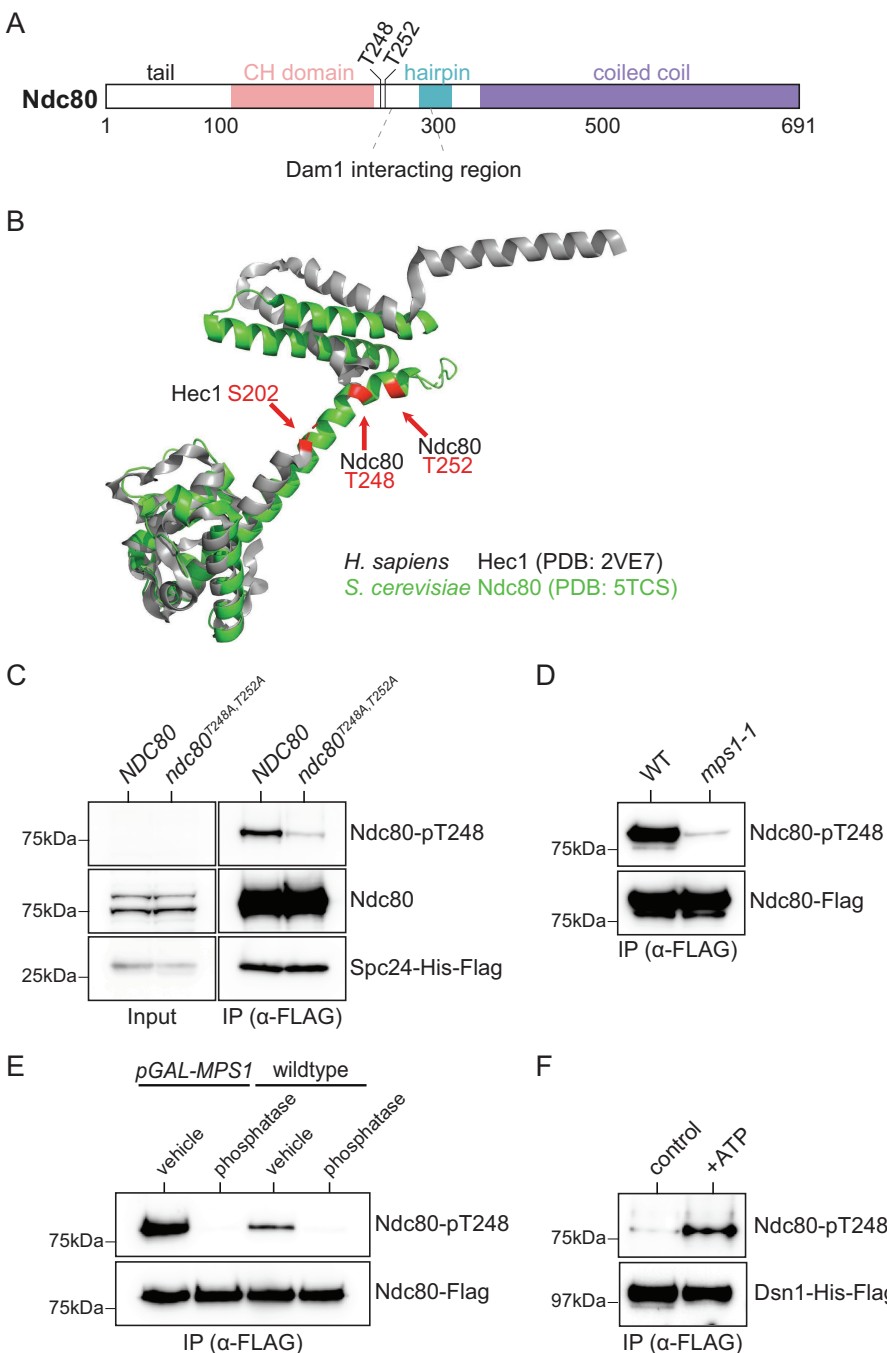

**Fig 2. Phosphorylation of Ndc80 at threonine-248 requires Mps1.** (A) Schematic of Ndc80. Thr-248 and Thr-252 are just C-terminal to the calponin-homology domain of Ndc80 and nearby the Dam1 interacting region. (B) Structural alignment of *H. sapiens* Hec1 (PDB: 2VE7; grey) with *S. cerevisiae* Ndc80 (PDB: 5TCS; green). Ndc80-T248 and Hec1-S202 are in similar regions of the corresponding structures. (C) Ndc80c was immunoprecipitated via Spc24-His-Flag from either *NDC80^WT^* (SBY21299) or *ndc80^T248A,T252A^* cells (SBY21301). Spc24-His-Flag (α-Flag), Ndc80 (α-Ndc80), and Ndc80 Thr-248 phosphorylation (α-Ndc80-pT248) were visualized via immunoblot. (D) Ndc80-Flag was immunoprecipitated from WT (SBY10651) and *mps1-1* cells (SBY20502) that were shifted to the non-permissive temperature 37 °C for 2.5 hours prior to harvest. Ndc80 phosphorylation at Thr-248 (α-Ndc80-pT248) and Ndc80-Flag (α-Flag) were visualized by immunoblot. (E) Ndc80-Flag was immunoprecipitated from WT (SBY10651) and *pGAL-MPS1* (SBY19721) cells grown in galactose for 2 hours. Immunoprecipitated proteins were treated with either lambda phosphatase or vehicle. Ndc80 phosphorylation at Thr-248 (α-Ndc80-pT248) and Ndc80-Flag (α-Flag) were

visualized by immunoblot. (F) Dsn1-His-Flag (SBY8253) was immunoprecipitated, washed, and then treated with either kinase buffer alone or kinase buffer with ATP. Dsn1-His-Flag (α-Flag) and Ndc80 Thr-248 phosphorylation (α-Ndc80-pT248) were visualized by immunoblot.

from the $ndc80^{T248A,T252A}$ purification, indicating specificity (Fig 2C). Next, we took multiple approaches to testing whether T248 is phosphorylated by Mps1. First, we purified Ndc80c from WT and *mps1–1* cells and found that the Ndc80 phospho-antibody signal was strongly reduced in *mps1–1* cells, consistent with Mps1 phosphorylating T248 *in vivo* (Fig 2D). Second, the Ndc80 complex (Ndc80-Flag) was immunoprecipitated from cells with and without Mps1 overexpression (*pGAL-MPS1*) and then treated with or without phosphatase (Fig 2E). The reactivity of the phospho-antibody against Ndc80 disappeared with phosphatase treatment and increased when Mps1 was overexpressed, consistent with an Mps1 phosphorylation site (Fig 2E). Finally, we tested whether Mps1 kinase could directly phosphorylate Ndc80 at T248 *in vitro*. To do this, we purified kinetochore particles (Dsn1-His-Flag) and treated the samples with ATP or buffer alone. We previously showed that Mps1 is the only kinase to co-purify with Dsn1-His-Flag kinetochore particles [21], and the ATP treatment dramatically increased T248 phospho-signal on immunoblots (Fig 2F). Together, these data, combined with the mass spectrometry results (Table 1), strongly suggest that Ndc80-T248 is an Mps1 phosphorylation site both *in vitro* and *in vivo*.

## Phosphorylation of Ndc80-T248 is cell cycle regulated

Next, we asked whether Ndc80-T248 phosphorylation is cell cycle regulated, which could indicate a role in chromosome segregation. We tested whether there was a difference in T248 phosphorylation between metaphase-arrested cells, using an auxin inducible degron of the anaphase promoting complex activator Cdc20 (*cdc20-AID*) [63], and anaphase-arrested cells, using the temperature sensitive *cdc15–2* (37 °C) mutant that prevents mitotic exit [64]. Asynchronous, *cdc20-AID*, and *cdc15–2* cells were grown at high temperature (37 °C) and treated with auxin to elicit arrest and then the Ndc80 complex (Ndc80-Flag) was purified. Although the levels of Ndc80 did not change, T248 phosphorylation was high in cycling and metaphase-arrested cells, but low in anaphase-arrested cells (Fig 3A). Together, these data suggest that Ndc80 is phosphorylated at T248 by Mps1 prior to or during metaphase and then dephosphorylated as cells enter anaphase.

Given that Ndc80-T248 phosphorylation peaks prior to anaphase (Fig 3A), we asked whether it is altered when microtubules are disrupted, which might suggest a role in error correction or the spindle assembly checkpoint. We arrested cells in metaphase (*cdc20-AID*) and treated them with or without nocodazole to depolymerize microtubules and to activate the spindle assembly checkpoint (Fig 3B). Ndc80-Flag was immunoprecipitated and we analyzed the phosphorylation state of T248 by immunoblotting with the phospho-specific antibody. As a control, we analyzed two other phosphorylation sites on the N-terminal tail of Ndc80, T54 and T74, that are error correction targets and phosphorylated upon nocodazole addition [28]. While there was an increase in T54 and T74 phosphorylation after nocodazole treatment, T248 phosphorylation was unchanged (Fig 3B). Next, we asked whether T248 phosphorylation occurs specifically in response to biorientation defects. Depletion of *KAR3* creates syntelic attachments which must be corrected and should therefore lead to increased phosphorylation of error correction targets such as the Ndc80 N-terminal tail [65]. To test this, we arrested cells in metaphase (*cdc20-AID*) with or without Kar3 (*kar3-AID*) and purified the Ndc80c (Ndc80-Flag). We found that similar to nocodazole treatment, Ndc80 T54 and T74 phosphorylation increased in *kar3-AID* cells but T248 phosphorylation remain unchanged (Fig 3C). Taken

together, these data suggest that the function of T248 phosphorylation is unlikely to be related to the spindle assembly checkpoint or error correction.

## Phosphorylation of Ndc80-T248 does not have detectable kinetochore functions

To analyze the function of the Ndc80 phosphorylation, we generated a phosphomimetic mutant, ndc80$^{T248D,T252D}$, at the endogenous locus in addition to the ndc80$^{T248A,T252A}$ phospho-deficient mutant. We verified that the mutations do not alter Ndc80 protein levels (S2A Fig). We tested whether there were any growth defects by performing a serial-dilution assay to assess growth at 23 °C and 37 °C, as well as sensitivity to the microtubule depolymerizing drug, benomyl. Both *ndc80*$^{T248A,T252A}$ and *ndc80*$^{T248D,T252D}$ grew similarly to WT cells, and didn't exhibit sensitivity to benomyl like the *mad3Δ* mutant control (Fig 4A). To more precisely determine whether Ndc80 T248/T252 phosphorylation affects cell cycle progression, we released *NDC80*$^{WT}$, *ndc80*$^{T248A,T252A}$ and *ndc80*$^{T248D,T252D}$ cells from G1 and analyzed the levels of the anaphase inhibitor Pds1 (Pds1-Myc) [66,67]. We found that Pds1 accumulation and degradation had similar kinetics in all three strains indicating that cell cycle progression is normal, and the spindle assembly checkpoint is not activated (Fig 4B).

We next asked if Ndc80 phosphorylation at T248/T252 alters the composition of kineto-chores. To do this, we purified Dsn1-His-Flag from asynchronously growing cells expressing *NDC80*$^{WT}$, *ndc80*$^{T248A,T252A}$ and *ndc80*$^{T248D,T252D}$ (S2B Fig). When we analyzed the purifications by MS and silver-stained PAGE, we did not detect any significant changes in protein abundance in either mutant, suggesting that Ndc80 phosphorylation does not alter kinetochore protein composition (Fig 4C). Since Ndc80-T248 is dephosphorylated in anaphase, we also purified kinetochores from *ndc80*$^{T248A,T252A}$ and *ndc80*$^{T248D,T252D}$ cells arrested in anaphase (*cdc15–2*) at 37 °C and analyzed them via mass spectrometry and silver-stained PAGE (S2C and S2D Fig). Like

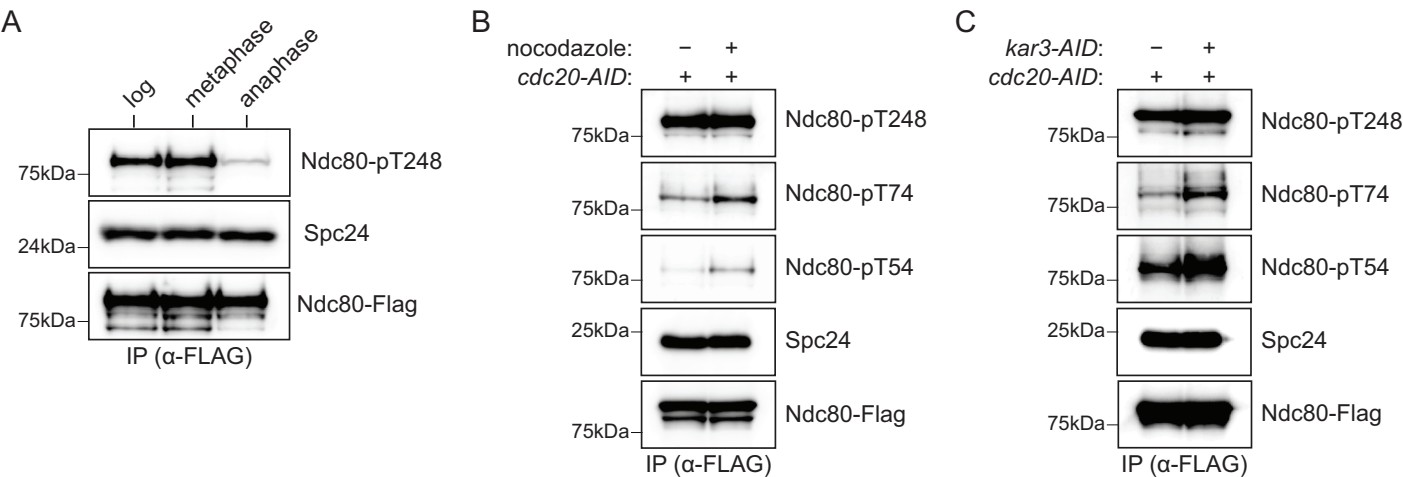

**Fig 3. Phosphorylation of Ndc80 Thr-248 is cell cycle regulated.** (A) Ndc80-Flag was immunoprecipitated from log phase (SBY10651), *cdc20-AID* (SBY21171, metaphase), and *cdc15-2* (SBY21544, anaphase) cells. All cells were shifted to 37 °C for 2.5 hours prior to harvest. *cdc20-AID* cells were treated with auxin just prior to temperature shift. Ndc80 phosphorylation at Thr-248 (α-Ndc80-pT248), Ndc80-Flag (α-Flag), and Spc24 (α-Spc24) was monitored by immunoblot. (B) Ndc80-Flag (SBY21171) cells were arrested in metaphase with auxin for 2.5 hours (*cdc20-AID*) and then treated with either nocodazole or vehicle for 30 minutes prior to harvest. Ndc80-Flag was immunoprecipitated. Ndc80-Flag (α-Flag), Spc24 (α-Spc24), and Ndc80 phosphorylation (α-Ndc80-pT54, α-Ndc80-pT74, and α-Ndc80-pT248) were analyzed by immunoblot. (C) Ndc80-Flag was immunoprecipitated from *cdc20-AID* (SBY21171) and *cdc20-AID kar3-AID* (SBY24251) cells. Cells were shifted to 37 °C for 2.5 hours prior to harvest. Ndc80-Flag (α-Flag), Spc24 (α-Spc24), and Ndc80 phosphorylation (α-Ndc80-pT54, α-Ndc80-pT74, and α-Ndc80-pT248) were analyzed by immunoblot.

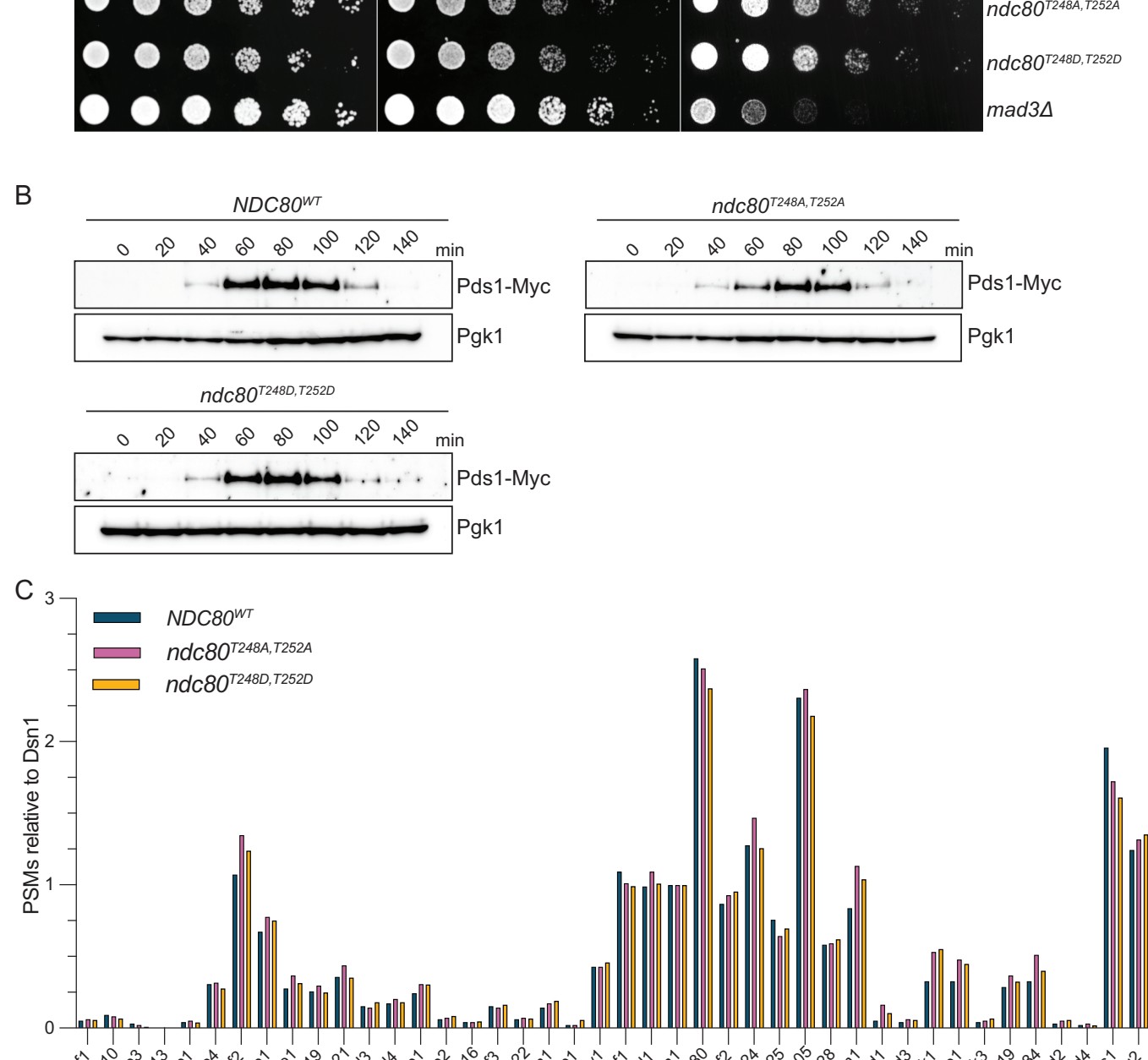

**Fig 4. Ndc80 phosphomutants have normal kinetochore composition.** (A) Viability and benomyl sensitivity of *NDC80*^WT^ (SBY21633), *ndc80*^T248A,T252A^ (SBY21725), *ndc80*^T248D,T252D^ (SBY21768), and *mad3Δ* (SBY293) yeast strains were tested via a serial dilution assay. (B) *NDC80*^WT^ (SBY22028), *ndc80*^T248A,T252A^ (SBY22030), and *ndc80*^T248D,T252D^ (SBY22029) cells were released from a G1 arrest and Pds1-Myc accumulation and degradation were monitored by immunoblot (α-Myc). Pgk1 (α-Pgk1) was analyzed via immunoblot as a loading control. (C) Mass spectrometry of kinetochore purifications (Dsn1-His-Flag IPs) from *NDC80*^WT^ (SBY21353), *ndc80*^T248A,T252A^ (SBY21287), and *ndc80*^T248D,T252D^ (SBY21289) genetic backgrounds. Raw peptide spectral match (PSM) counts are presented relative to Dsn1 ("relative PSMs") for co-purifying kinetochore proteins.

the asynchronous cell purifications, no dramatic changes in kinetochore composition were apparent. We also tested whether the phosphomutants alter the association of Ndc80 with kinetochores *in vivo*. We quantified the ratio of Nuf2-mGFP (an Ndc80c member) to Mtw1-mKate (a fiducial kinetochore marker) in both metaphase and anaphase cells. There was not a significant change in Nuf2-mGFP signal in either *ndc80*$^{T248A,T252A}$ or *ndc80*$^{T248D,T252D}$ cells (S3A Fig) and we didn't notice any other changes in Nuf2-mGFP localization in either genotype (S3B and S3C Fig). Taken together, these data indicate that Ndc80 phosphorylation at T248 is removed at anaphase onset, but that this phospho-event is unlikely to play a key role in error correction, kinetochore assembly or microtubule attachment.

## Ndc80 phosphorylation regulates its interaction with the Dam1c

Because we did not detect any kinetochore function for the Ndc80 phosphorylation, we asked whether it might affect another activity. Earlier structural and cross-linking mass spectrometry studies found Dam1c binding in proximity to Ndc80 T248 and T252 (Fig 2A) [36–38,45,59]. We therefore asked whether Ndc80 phosphorylation affects the interaction between the total cellular pool of Ndc80 and Dam1c. First, we asked if their interaction changes between meta-phase and anaphase. We purified Ndc80c via Spc24-His-Flag from cells arrested in metaphase (*cdc20-AID*) and anaphase (*cdc15–2*) at 37 °C and analyzed the levels of co-purifying Dam1 complex using antibodies against Ask1, a Dam1c component. We found a reduction in Ask1 levels that co-purify with Ndc80 in anaphase-arrested cells (Fig 5A), suggesting that cells actively decrease the Ndc80c-Dam1c association at anaphase onset. To determine whether this change is regulated by Ndc80 phosphorylation, we asked whether the persistent presence of Ndc80-T248D-T252D in anaphase would enhance the Ndc80c-Dam1c interaction. We purified Ndc80c (Spc24-His-Flag) from *NDC80*$^{WT}$, *ndc80*$^{T248A,T252A}$ and *ndc80*$^{T248D,T252D}$ cells arrested in anaphase at 37 °C (*cdc15–2*). We found that the Ndc80-T248D-T252D complex retained more Ask1 protein than wildtype, whereas Ndc80-T248A-T252A co-purified similar amounts of Dam1c to wildtype (Fig 5B).

Although our biochemical purifications suggested that Ndc80 phosphorylation does not regulate the interaction between Ndc80 and Dam1 on kinetochores, previous studies showed that Dam1 kinetochore association decreases by about half in anaphase [68]. We therefore quantified Dam1c anaphase kinetochore localization *in vivo* in the *ndc80* phosphomutants. We released *NDC80*$^{WT}$, *ndc80*$^{T248A,T252A}$ and *ndc80*$^{T248D,T252D}$ cells expressing Dad1-GFP and Ndc10-mCherry (a fiducial kinetochore marker) from a G1 arrest into the cell cycle and analyzed anaphase kinetochores 60–90 minutes after release (Figs 5C, 5D, and S4). There was no statistically significant difference in the ratio of Dad1-GFP:Ndc10-mCherry signal between the three genotypes, indicating that Ndc80 phosphorylation on T248/252 is unlikely to be important for Dam1c kinetochore localization in anaphase (Fig 5D). Taken together, our data suggest that Ndc80 phosphorylation may regulate a pool of Dam1 that is not on kinetochores.

To determine what cellular population of Ndc80 and Dam1 is regulated by phosphorylation, we abolished kinetochore assembly using the *ndc10–1* mutant to test whether there is a soluble pool of Ndc80 and Dam1 that interact in a phospho-dependent manner [69]. We purified Ndc80c via Spc24-His-Flag from anaphase-arrested cells (*cdc15–2*) with and without kineto-chores (*ndc10–1*) in both the *NDC80*$^{WT}$ and *ndc80*$^{T248D,T252D}$ backgrounds. Cells were arrested in G1, released at 37 °C to ensure they went through replication in the absence of Ndc10, and then harvested when they were large-budded [69]. We found that Ask1 still co-purifies with Ndc80 in *ndc10–1* cells and there is only a slight reduction in the *ndc10–1* mutant, indicating that the majority of the Ndc80/Dam1 complex is not interacting at kinetochores (Fig 5E). Strikingly, the Ndc80-T248D-T252D interacted with more Ask1 protein than wildtype Ndc80 even in cells

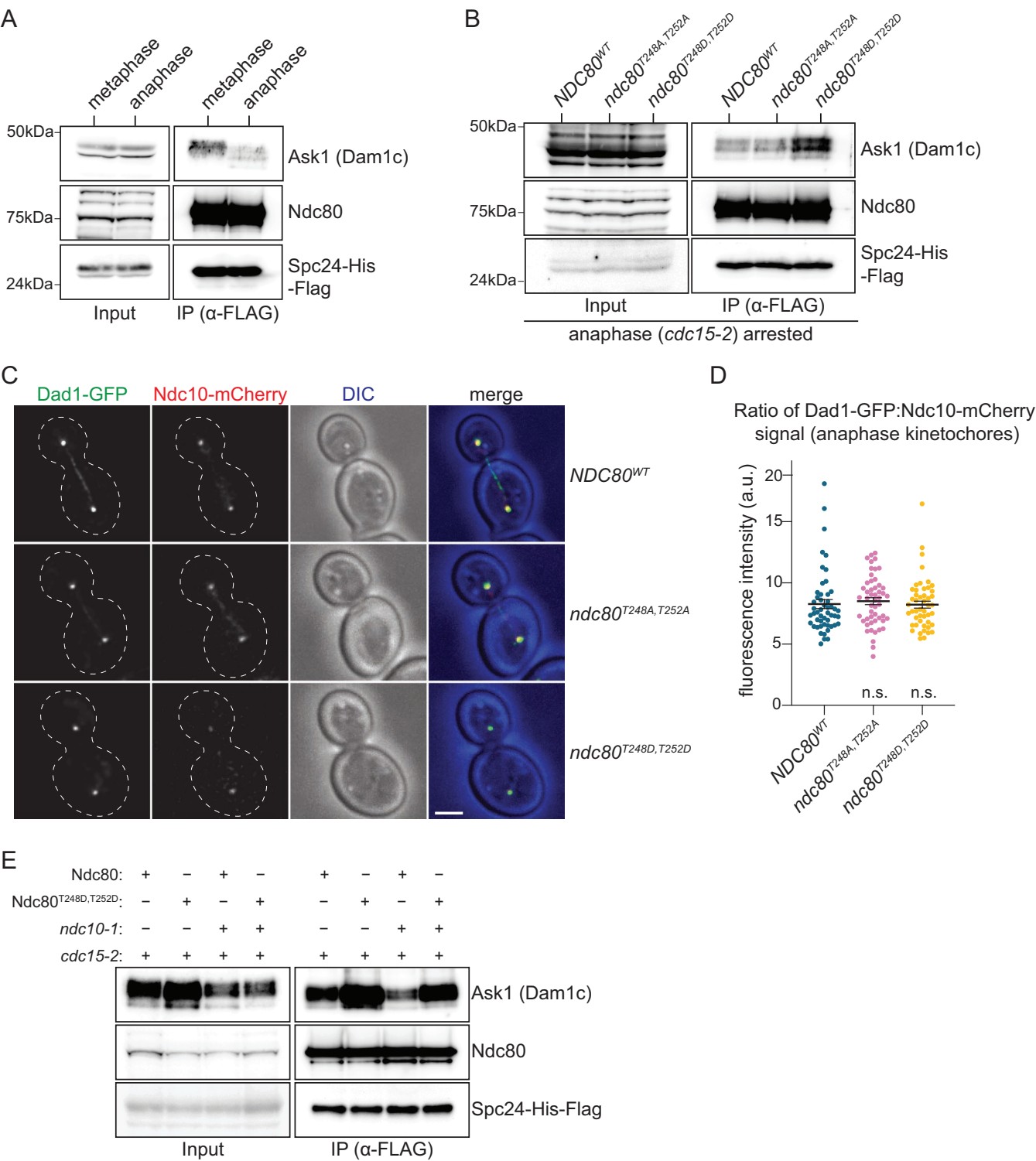

**Fig 5. Ndc80 phosphorylation regulates its interaction with Dam1c.** (A) Ndc80c was immunoprecipitated via Spc24-His-Flag from *cdc20-AID* (metaphase; SBY19699) and *cdc15-2* (anaphase; SBY19701) arrested cells. Cells were shifted to 37 °C for 2.5 hours prior to harvest. Spc24-His-Flag (α-Flag), Ndc80 (α-Ndc80), and Ask1 (α-Ask1) levels were analyzed via immunoblot. (B) Ndc80c was immunoprecipitated via Spc24-His-Flag from *cdc15-2* (anaphase) arrested *NDC80^{WT}* (SBY22551), *ndc80^{T248A,T252A}* (SBY22554), and *ndc80^{T248D,T252D}* (SBY22557) cells. Cells were shifted to 37 °C for 2.5 hours prior to harvest. Spc24-His-Flag (α-Flag), Ndc80 (α-Ndc80), and Ask1 (α-Ask1) levels were analyzed via immunoblot. (C) Representative images of *NDC80^{WT}* (SBY23673), *ndc80^{T248A,T252A}* (SBY23675), and *ndc80^{T248D,T252D}* (SBY23677) cells in anaphase expressing Dad1-GFP and Ndc10-mCherry. Scale bar represents 2 µm. (D) Anaphase kinetochores were analyzed 60-90 minutes after G1 release in *NDC80^{WT}* (SBY23673), *ndc80^{T248A,T252A}* (SBY23675), and *ndc80^{T248D,T252D}* (SBY23677)

cells expressing Dad1-GFP and Ndc10-mCherry. Quantification of anaphase kinetochore signal is shown as the ratio of Dad1-GFP:Ndc10-mCherry signal across 51 kinetochore clusters. Each point represents a kinetochore pair. Error bars represent SEM centered around the mean. n.s. indicates not significant compared to $NDC80^{WT}$ ($ndc80^{T248A,T252A}$ p-value >0.61; $ndc80^{T248D,T252D}$ p-value > 0.91). Mean values: $NDC80^{WT}$= 8.27±1.16, $ndc80^{T248A,T252A}$ = 8.51±1.19, $ndc80^{T248D,T252D}$ = 8.22±1.15. (E) Ndc80c was immunoprecipitated via Spc24-His-Flag from $cdc15-2$ (SBY22551), $ndc80^{T248D,T252D}$ $cdc15-2$ (SBY22557), $cdc15-2$ $ndc10-1$ (SBY24281), and $ndc80^{T248D,T252D}$ $cdc15-2$ $ndc10-1$ (SBY24282) cells. Cells were arrested with alpha-factor, released at 37 °C, and large-budded cells were harvested. Spc24-His-Flag (α-Flag), Ndc80 (α-Ndc80), and Ask1 (α-Ask1) levels were analyzed via immunoblot.

lacking kinetochores ($ndc10–1$), indicating that the major role of Ndc80 phosphorylation is to promote the interaction of soluble Ndc80/Dam1 complexes (Fig 5E).

## Ndc80 phosphorylation regulates the levels of Dam1c on spindles

Although Ndc80c does not localize to spindles, a subset of Dam1c localizes to the spindle in mitosis and $dam1$ mutants have defective spindles in addition to kinetochore-microtubule attachment defects [48,50,70,71]. We therefore considered the possibility that the pool of Dam1 that associates with spindles is regulated through its interaction with Ndc80. However, to our knowledge, Dam1 behavior on spindles throughout mitosis has not been fully described, so we analyzed Damc1 localization during a wildtype mitosis in cells expressing Dad1-mKate2 and GFP-Tub1. Cells were released from a G1 arrest and then fixed and imaged every 20 minutes (S5A Fig). In addition to its robust kinetochore localization throughout mitosis, Dam1 exhibits weaker localization to most metaphase spindles (S4A Fig; 80' and 100'). At anaphase, Dam1c forms a punctate pattern along approximately two-thirds of spindles (S5A Fig; 120'). We note that the presence of Dam1c on anaphase spindles does not appear to correlate with spindle length, i.e., there were plenty of long spindles, beginning to break down, where Dad1 was still present (S5B Fig).

We next tested whether the phospho-status of Ndc80 at T248/T252 affected Dam1c localization to the mitotic spindle. We were unable to accurately quantify Dam1c levels specifically on metaphase spindles because the spindles were so short that it was difficult to distinguish the spindle pool from the kinetochore-bound complex. We therefore analyzed anaphase spindles by arresting $NDC80^{WT}$, $ndc80^{T248A,T252A}$ and $ndc80^{T248D,T252D}$ cells expressing Dad1-GFP and Mtw1-mCherry in G1 and releasing them into the cell cycle. We qualitatively analyzed anaphase spindles after release for the presence or absence of Dad1-GFP (Fig 6A). We found that the fraction of $ndc80^{T248D,T252D}$ anaphase spindles containing Dad1-GFP was significantly reduced, with only approximately 45% localizing Dad1-GFP (Fig 6B; p-value <0.01), compared to 63% for $NDC80^{WT}$ and 70% for $ndc80^{T248A,T252A}$. Next, we quantitatively analyzed the levels of Dam1c on anaphase spindles using Dad1-mKate2, with an ectopic copy of GFP-Tub1 serving as a fiducial marker, utilizing anaphase cells from log-phase cultures (Figs 6C, 6D, S5C and S5D). However, GFP-Tub1 incorporation into $ndc80^{T248D,T252D}$ spindles was significantly reduced compared to $NDC80^{WT}$ and $ndc80^{T248A,T252A}$ spindles, so we did not quantify Dad1 levels for these spindles although we note the overall Dad1 signal intensity was down on $ndc80^{T248D,T252D}$ spindles (S5C and S5D Fig). We were able to quantify $NDC80^{WT}$ and $ndc80^{T248A,T252A}$ cells and found that the $ndc80^{T248A,T252A}$ cells had an increased Dad1-mKate2:GFP-Tub1 ratio on anaphase spindles when compared to $NDC80^{WT}$ (mean values: $ndc80^{T248A,T252A}$ = 0.100 vs $NDC80^{WT}$ = 0.084; p-value <0.02; Fig 6D). This difference was not dependent on spindle length, as there was no statistically significant difference in the lengths of the spindles in our dataset between genotypes (S5E Fig). This result is consistent with our qualitative analysis (Fig 6B) and indicates that Ndc80 dephosphorylation may promote Dam1c localization to anaphase spindles.

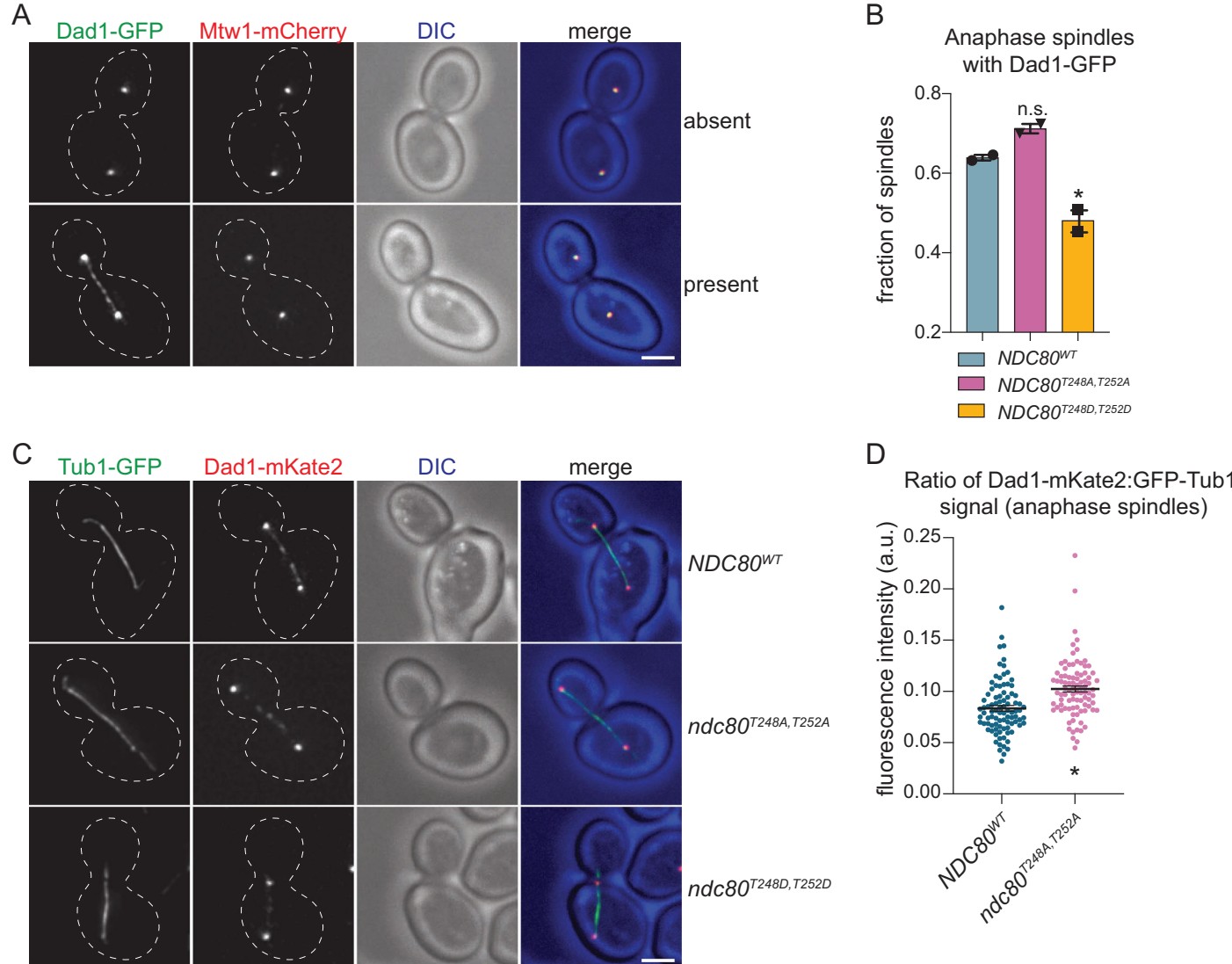

**Fig 6. *ndc80* phospho-mutants have altered Dam1c levels on anaphase spindles.** (A) Anaphase spindles were analyzed 60–90 minutes after G1 release in *NDC80^{WT}* (SBY23361), *ndc80^{T248A,T252A}* (SBY23380), and *ndc80^{T248D,T252D}* (SBY23385) cells expressing Dad1-GFP and Mtw1-mCherry. Representative images of anaphase spindles are shown with and without Dad1-GFP on the spindle. Scale bar represents 2 μm. (B) Quantification of anaphase spindles from (A). Each individual experiment (n>200 anaphase spindles) is represented as a symbol above the bar graph. Error bars represent SEM. * indicates p-value <0.009 compared to *NDC80^{WT}*. n.s. indicates not significant compared to *NDC80^{WT}* (p-value > 0.13). Mean values: *NDC80^{WT}* = 63%±4, *ndc80^{T248A,T252A}* = 70%±5, *ndc80^{T248D,T252D}* = 45%±3. (C) Anaphase spindles were analyzed in *NDC80^{WT}* (SBY23144), *ndc80^{T248A,T252A}* (SBY23145), and *ndc80^{T248D,T252D}* (SBY23146) cells expressing GFP-Tub1 and Dad1-mKate2. A representative image of an anaphase spindle from each genotype is shown. Scale bar represents 2 μm. (D) Quantification of anaphase spindles analyzed in (C). Quantification of spindle intensity of Dam1c is shown as the ratio of Dad1-mKate:GFP-Tub1 across 75 anaphase spindles that had detectable Dad1 fluorescence (3 biological replicates). Error bars represent SEM centered around the mean. * indicates p-value <0.02 compared to *NDC80^{WT}*. Mean values: *NDC80^{WT}* = 0.084±0.01, *ndc80^{T248A,T252A}* = 0.100±.01.

## *ndc80* phosphomutants exhibit genetic interactions and altered spindles with *dam1* mutants

Because our work suggested that Dam1c levels on spindles are altered by Ndc80 phosphorylation, we next tested whether the *ndc80* phospho-mutants exhibit genetic interactions with Dam1c mutations. We crossed the *ndc80* phospho-alleles to the Dam1c temperature sensitive mutants *ask1–2* and *dad1–1* [56,71] and performed serial dilution assays at 23 °C, 33 °C

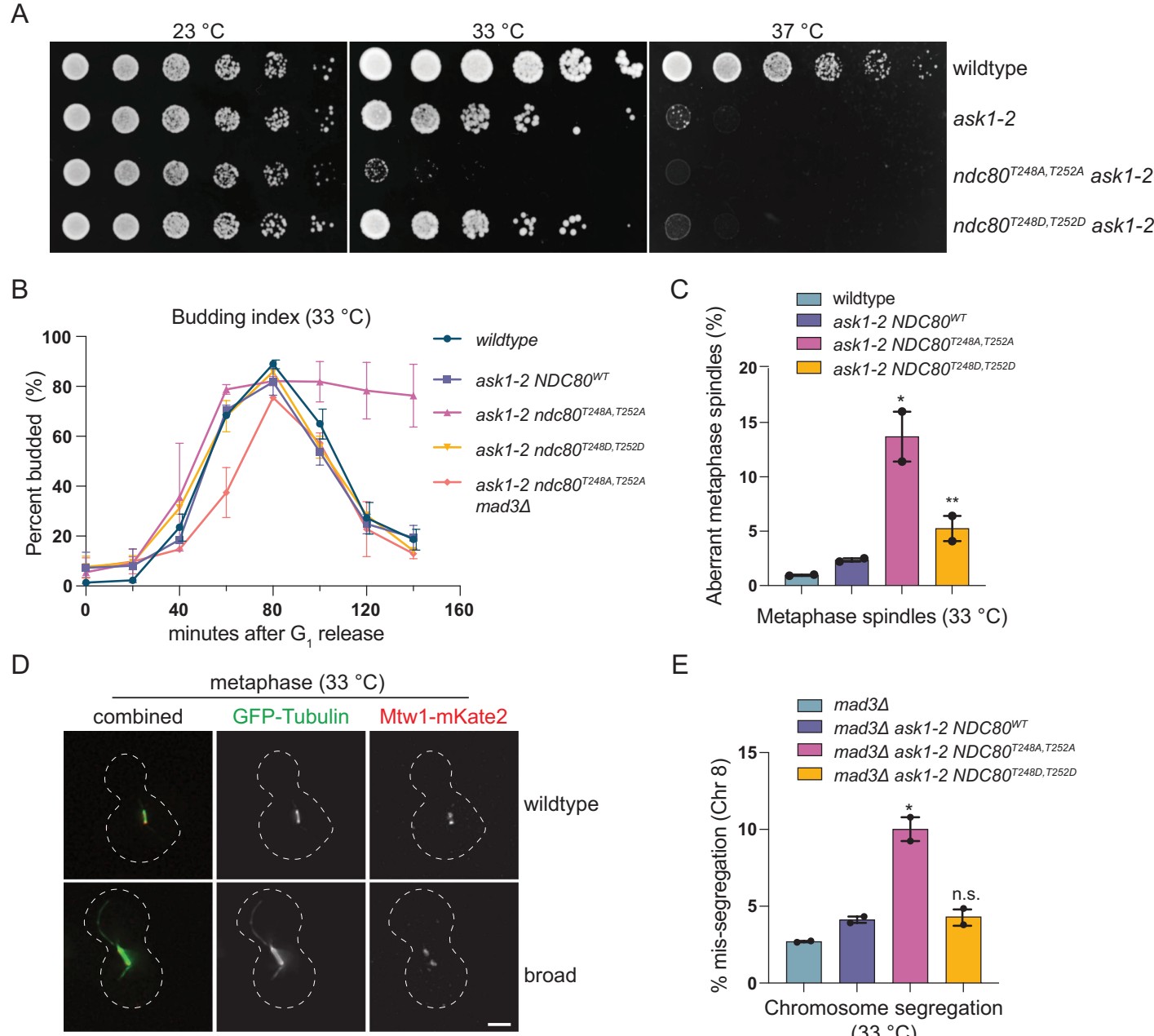

**Fig 7. ndc80$^{T248A,T252A}$ genetically interacts with dam1 temperature sensitive mutants and causes spindle defects.** (A) Serial dilutions of WT (SBY3), *ask1–2* (SBY2730), *ndc80*$^{T248A,T252A}$ *ask1–2* (SBY22732), and *ndc80*$^{T248D,T252D}$ *ask1–2* (SBY22734) were plated on YPD and grown at the specified temperatures. (B) Budding index (% budded cells) was measured in WT (SBY3), *ask1–2* (SBY2730), *ndc80*$^{T248A,T252A}$ *ask1–2* (SBY22732), *ndc80*$^{T248D,T252D}$ *ask1–2* (SBY22734), and *ndc80*$^{T248A,T252A}$ *ask1–2 mad3Δ* (SBY24257) cells. Cells were arrested in G$_1$ with alpha-factor, released at 33 °C, and cell cycle progression was monitored every 20 minutes. The mean (% budded) from two independent experiments (n>200 cells for each timepoint) is graphed with error bars representing SEM. (C) WT (SBY22771), *ask1–2* (SBY22852), *ndc80*$^{T248A,T252A}$ *ask1–2* (SBY22853), and *ndc80*$^{T248D,T252D}$ *ask1–2* (SBY22854) cells, all containing GFP-Tub1 and Mtw1-mKate2 were released from G1 after alpha-factor treatment and metaphase spindles were imaged and analyzed as cells progressed through mitosis. Each individual experiment (n>200 metaphase spindles) is represented as a symbol above the bar graph. Error bars represent SEM. * indicates p-value <0.0001, ** indicates p-value >0.03, both compared to *NDC80*$^{WT}$ *ask1–2*. Mean values: WT = 1.0%±0.1, *NDC80*$^{WT}$ *ask1–2* = 2.3%±0.1, *ndc80*$^{T248A,T252A}$ *ask1–2* = 13.7%±2.3, *ndc80*$^{T248D,T252D}$ *ask1–2* = 5.2%±1.1. (D) Representative micrographs are shown of WT and mutant ("broad") metaphase spindles from quantification in (C). Scale bar represents 2 μm. (E) Mis-segregation of chromosome 8 was measured using a fluorescently labeled CEN8 signal in *mad3Δ* (SBY24290), *ask1–2 mad3Δ* (SBY24184), *ndc80*$^{T248A,T252A}$ *ask1–2 mad3Δ* (SBY24185), and *ndc80*$^{T248D,T252D}$ *ask1–2 mad3Δ* (SBY24186) cells. The mean from two independent experiments (n>200 cells for each genotype) is graphed with error bars representing SEM and individual experimental values shown as symbols on the bar graphs. * indicates p-value <0.02 compared to *ask1–2 mad3Δ*. Mean values: *mad3Δ* = 2.7%, *ask1–2 mad3Δ* = 4.1%, *ask1–2 mad3Δ ndc80*$^{T248A,T252A}$ =10.0%, *ask1–2 mad3Δ ndc80*$^{T248D,T252D}$ = 4.3%.

(semi-permissive temperature) and 37 °C (restrictive temperature). While *ndc80*<sup>*T248D,T252D*</sup> did not alter the growth of either *dad1–1* or *ask1–2* at these temperatures, *ndc80*<sup>*T248A,T252A*</sup> exhibited a strong genetic interaction with both alleles (Figs 7A and S6A). At 33 °C, the *ndc80*<sup>*T248A,T252A*</sup> *dad1–1* and *ndc80*<sup>*T248A,T252A*</sup> *ask1–2* double mutants were very slow growing, whereas the single mutants were nearly fully viable at this temperature. Thus, although Ndc80 T248/T252 phosphorylation is non-essential, it becomes critical when Dam1 complex function is crippled.

Next, we sought to determine the cause of the *ndc80*<sup>*T248A,T252A*</sup> *ask1–2* synthetic interaction at the semi-permissive temperature. To this end, we performed a budding index assay in WT, *ask1–2 NDC80*<sup>*WT*</sup>, *ask1–2 ndc80*<sup>*T248A,T252A*</sup>, *ask1–2 ndc80*<sup>*T248D,T252D*</sup>, and *ask1–2 ndc80*<sup>*T248A,T252A*</sup> *mad3Δ* cells. Cells were arrested in G1, released into the cell cycle at 33 °C, and the percent of budded cells for each genotype was counted every 20 minutes. While *ask1–2 NDC80*<sup>*WT*</sup> and *ask1–2 ndc80*<sup>*T248D,T252D*</sup> cells progressed through the cell cycle with similar kinetics to WT cells, *ask1–2 ndc80*<sup>*T248A,T252A*</sup> cells delayed as large-budded cells (Fig 7B). Furthermore, this delay was dependent on the spindle assembly checkpoint, as *ask1–2 ndc80*<sup>*T248A,T252A*</sup> *mad3Δ* cells passed through the cell cycle similarly to WT. These data indicate that the *ask1–2 ndc80*<sup>*T248A,T252A*</sup> synthetic interaction is due to a mitotic defect that triggers the spindle assembly checkpoint.

To further explore the *ask1–2 ndc80*<sup>*T248A,T252A*</sup> genetic interaction, we analyzed kinetochores and spindles in mitosis. Cells containing fluorescently labeled tubulin (GFP-Tub1) and kinetochores (Mtw1-mKate2) were released from G1 to the semi-permissive temperature (33 °C) and both metaphase and anaphase cells were analyzed after release. Mtw1-mKate2 appeared similar in all strains, with most cells bi-orienting clustered kinetochores by 60'. However, when we analyzed spindles, we found that *ndc80*<sup>*T248A,T252A*</sup> *ask1–2* metaphase spindles were atypically broad and wide (Fig 7C; 13.7%; p-value <0.0001), appearing to incorporate additional tubulin and sometimes giving a "fish-hook" like appearance (Fig 7D)[49]. In addition, the population of *ndc80*<sup>*T248A,T252A*</sup> *ask1–2* cells that were able to enter anaphase were frequently wavy or curved and exhibited inconsistent GFP-Tub1 incorporation (S6C Fig). While these phenotypes were detected in the single mutants, they were significantly enriched in *ndc80*<sup>*T248A,T252A*</sup> double mutants (S6B and S6C Fig; 16.8%; p-value <0.0001). This is consistent with the *ndc80*<sup>*T248A,T252A*</sup> *ask1–2* synthetic interaction being caused, at least in part, by anomalous mitotic spindles. Furthermore, it supports the model that Ndc80 phosphorylation at T248/T252 regulates the function of Dam1c in forming proper mitotic spindles to ensure faithful chromosome segregation.

The defective spindle morphology and activation of the spindle checkpoint suggested that the *ask1–2 ndc80*<sup>*T248A,T252A*</sup> cells have mitotic defects that could lead to chromosome mis-segregation. To directly analyze this, we utilized the LacI-GFP system with *LacO* integrated at the centromere of chromosome 8 (Chr8) to mark a single chromosome and monitor its segregation [72]. All cells had a *MAD3* deletion to ensure cell cycle progression. Cells were released from G1 to 33 °C and segregation of Chr8 was quantified in anaphase cells. *ndc80*<sup>*T248A,T252A*</sup> *ask1–2 mad3Δ* cells had a significant increase in mis-segregation compared to *NDC80*<sup>*WT*</sup> *ask1–2 mad3Δ* cells (Fig 7E; p-value <0.02). Taken together, our data suggests that regulation of the Ndc80-Dam1 interaction by phosphorylation on T248/T252 becomes important for mitotic fidelity when the Dam1 complex is crippled.

## Discussion

Here we report a phospho-proteomic analysis of native yeast kinetochores and identify many previously unknown sites that may have important functional roles in mitosis. We focused on the conserved Mps1 kinase and discovered 14 previously unstudied potential Mps1-mediated phosphorylation events on kinetochore proteins. We confirmed that Ndc80 is phosphorylated *in vivo* by Mps1 on T248 and possibly T252, residues that are near the Dam1c interaction site. Consistent with this, we found that Mps1 phosphorylation contributes to the interaction

between Ndc80 and Dam1, which in turn affects the pool of Dam1 that localizes to and regulates the mitotic spindle.

Our proteomic analysis identified 37 kinetochore phosphorylation sites matching the Mps1, Ipl1, Cdk1, or Cdc5 consensus motif that have not been previously reported on the Saccharomyces Genome Database (SGD; S2 Table). For example, we identified many sites on the Ndc80c. There were seven phosphorylation events detected on Spc24, one on Spc25, two on Nuf2, and thirteen on the Ndc80 protein. Seven of these matched the Mps1 consensus motif, nine matched the Ipl1 motif, and two matched the Cdk1 motif, indicating robust regulation of Ndc80c by numerous kinases. However, we did not identify some previously described phosphorylation sites (such as phosphorylation of several MELT motifs on Spc105)[21], indicating that there are many more sites that were not detected in our dataset. Additionally, we detected 25 phospho-sites that are not reported in SGD and that didn't map to any of the kinase motifs we searched for (S2 Table). We focused on Mps1 because its essential error correction targets are not known. In yeast, the Ndc80 and Dam1 microtubule binding complexes are the key substrates for Aurora B's error-correction function, and it is possible the same sites are regulated by Mps1, or that there are redundancies between known Mps1 sites that haven't been tested. However, we identified many additional potential Mps1 sites in Ndc80 and Dam1 that could be involved in error correction (S1 Table). Because most of these phospho-sites map to unstructured or flexible protein regions, it is not possible to predict whether they would affect error correction and thus they will have to be tested individually. We also identified Mps1 sites that are likely involved in kinetochore functions unrelated to error correction. For example, we detected several phosphorylation sites near the C-terminus of Dsn1 (i.e., S546, S547, S554) that may be Mps1 (or Cdc5) regulated and need to be tested for a role in chromosome segregation. Although we did not determine whether any of the phosphorylation events are cell cycle regulated, the phosphorylation we focused on (Ndc80-T248) is removed at anaphase onset. In the future, determining which events are regulated during mitosis may also shed light on their functions.

Our work suggests that phosphorylation of Ndc80 at T248/252 regulates the interaction between the Ndc80 and Dam1 complexes. Consistent with Mps1 activity decreasing at anaphase [73], the Ndc80 T248 phosphorylation was reversed at anaphase and the interaction between Ndc80 and Dam1 decreased at anaphase. Phosphorylation of Ndc80 T248/252 regulates this association because a $ndc80^{T248D,T252D}$ mutant enhanced the Ndc80/Dam1 interaction in anaphase-arrested cells. We did not detect changes in the amount of Dam1 that purified with kinetochores in cells expressing $ndc80^{T248D,T252D}$, suggesting that the phosphorylation does not directly regulate the kinetochore-bound pool. Consistent with this, we did not detect changes in Dam1 levels on kinetochores in cells and we found that the phosphorylation promoted the interaction between Ndc80c and Dam1c in cells lacking kinetochores. We therefore propose that Ndc80 phosphorylation regulates the soluble pool of Dam1 to control the levels of free Dam1 that are available to localize to the spindle (Fig 8). However, additional regulatory events also contribute to Dam1c spindle localization since a portion of $ndc80^{T248D,T252D}$ cells still localized Dam1c to anaphase spindles. In the future, it will be important to identify the phosphatase that controls the Ndc80-Dam1c interaction at the metaphase to anaphase transition to ensure spindle integrity. The Ndc80 phosphorylation events we identified might be conserved because the human Hec1 protein is phosphorylated on S202 [74], a site which matches the Mps1 consensus motif and is in similar region of the protein, C-terminal to the CH domain. Although the Dam1 complex is not conserved in humans, the Ska complex is considered to be a functional ortholog [75], so it will be important to determine whether Hec1 phosphorylation on these sites regulates the interaction between Hec1 and the Ska complex.

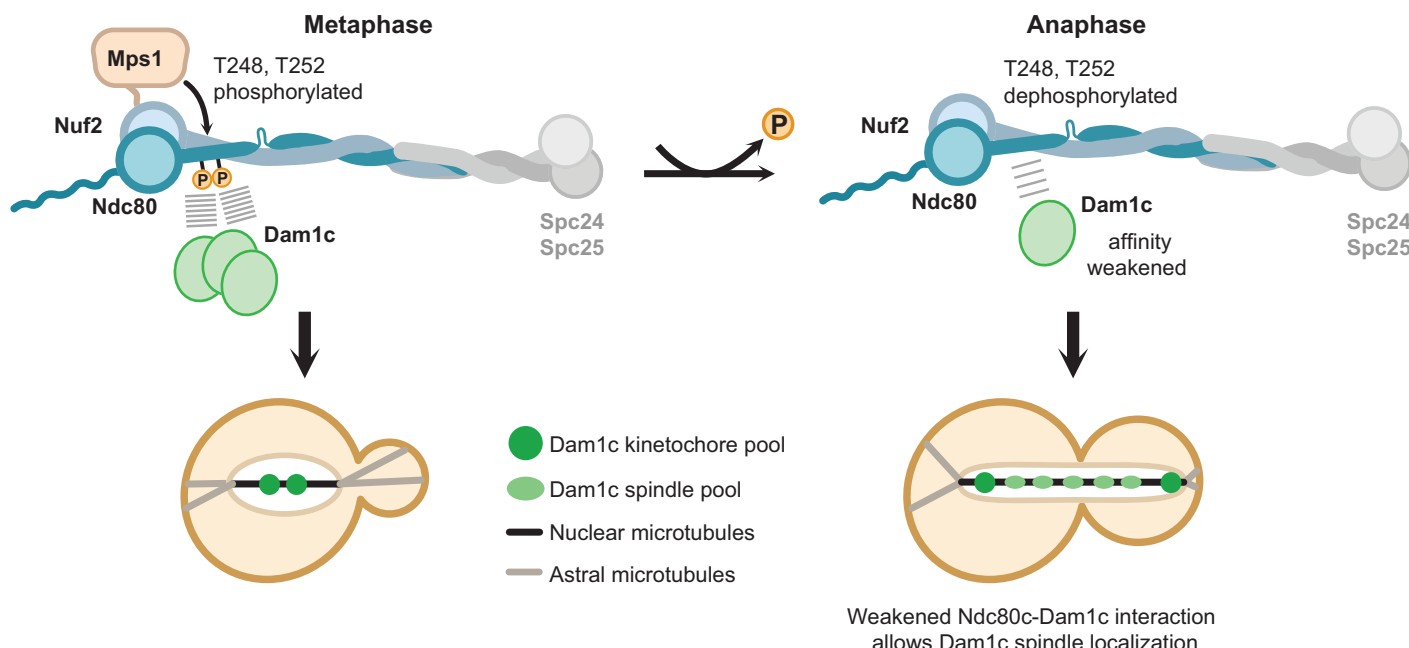

**Fig 8. Model: Ndc80 phosphorylation is relieved in anaphase to promote Dam1c localization to the spindle.** Mps1 phosphorylates Ndc80 at Thr-248 and Thr-252, peaking in metaphase. In metaphase, the majority of Dam1c localizes to clustered kinetochores. As cells enter anaphase, phosphorylation of Ndc80 is removed and a pool of Dam1c exhibits increased spindle localization. We propose that dephosphorylation of Ndc80 at Thr-248/Thr-252 contributes to the spindle localization of Dam1c in anaphase, perhaps via weakening of the Ndc80-Dam1 interaction.

The precise molecular function of Dam1c on spindles is not known but it can promote microtubule assembly and bundle microtubules *in vitro* [42]. These functions may be related to its ability to oligomerize around the microtubule [34,42,76–79], but it hasn't been clear whether it directly regulates microtubule dynamics *in vivo*. Although *dam1* mutants exhibit spindle defects *in vivo* [48,50,70,71], these mutants also affect kinetochore-microtubule attachments. It has therefore been difficult to determine the function of Dam1c in regulating spindles. The mutants in the Ndc80 phospho-sites that altered the interaction with Dam1c did not have phenotypes on their own, so additional mechanisms ensure accurate spindle function. However, we detected major defects in both mitotic spindles when the Dam1 complex was crippled in the presence of *ndc80^T248A,T252A*, a mutant that also increased Dam1c levels on spindles. It isn't clear why the Ndc80 phosphomimetic mutant did not exhibit a genetic interaction with the *dam1* mutants, but we did notice a decrease in the incorporation of ectopic GFP-Tub1 incorporation into spindles in the *ndc80^T248D,T252D* cells (S5C Fig), suggesting there is also a problem with spindle integrity in this mutant. Taken together, these data strongly suggest that Dam1c regulates spindle integrity and that this function requires both the proper levels and function of Dam1c on microtubules. In the future, it will be important to determine whether Dam1c directly regulates spindle integrity or functions through interacting partners.

Cells appear to exert multiple phospho-regulatory controls over the Dam1 complex at the metaphase to anaphase transition. Cdk1 phosphorylation of the Dam1c promotes its oligomerization to strengthen kinetochore-microtubule interactions [56,57]. Furthermore, Dam1 is phosphorylated by Protein Kinase A to promote Dam1c stability and kinetochore localization [80]. Here, we found that the soluble pools of Ndc80 and Dam1 also appear to be phospho-regulated to control the amount of Dam1 available to load onto the mitotic spindle. Because Dam1 localizes to both kinetochores and microtubules, it is not surprising the cell has

evolved mechanisms to control the amount of complex available for spindle association. There are other kinetochore proteins that also localize to microtubules, such as the Stu2 microtubule associated protein and the Sli15 chromosomal passenger protein. Sli15 spindle localization is regulated by Cdk1 phosphorylation that prevents its interaction with microtubules [81,82], highlighting the conservation of phosphorylation as a key event to control the relative pools. In the future, the additional phosphorylation sites we identified on kinetochore proteins may shed light on other regulatory events that ensure accurate chromosome segregation.

## Materials & methods

### Yeast strain and plasmid construction

All *Saccharomyces cerevisiae* strains in this study are derived from the W303 genetic background (SBY3) and complete genotypes are listed in S3 Table. All strains were constructed using standard genetic methods [83]. Plasmids used in this study are available in S4 Table and primer sequences are in S5 Table. Construction of *DSN1–6His-3FLAG* is described in [52], *SPC24–6His-3FLAG* in [53], *GFP-TUB1:LEU2* in [84], and *NDC80–3FLAG* and *DAD1–3FLAG* in [85]. *Ndc80-T248A-T252A:HPHMx* and *ndc80-T248D-T252D:HPHMx* were constructed by first inserting the *HPHMx* cassette 200 basepairs 3' to the stop codon of *NDC80* using homologous recombination [86]. Mutations were then made in *NDC80* using a single-step CRISPR/Cas9 edit as previously described [87]. *NDC10-mCherry*, *MTW1-mKate2*, *DAD1-GFP*, *DAD1-mKate2*, and *NUF2-mGFP* were constructed using PCR amplification and homologous recombination techniques [88,89]. Yeast containing the *cdc20-AID* allele were a gift from Adele Marston's laboratory (Wellcome Center for Cell Biology, University of Edinburgh, Edinburgh, UK). *pGAL-MPS1-MYC:URA3* yeast were a gift from Kevin Hardwick's laboratory (School of Biological Sciences, University of Edinburgh, Edinburgh, UK). Yeast containing the *mps1–1* allele were a gift from Andrew Murray's laboratory (Harvard University, Cambridge, MA). *cdc15–2* yeast were a gift from David Morgan's laboratory (University of California, San Francisco, CA). *dad1–1:KAN* yeast were a gift from the lab of Georjana Barnes (University of California, Berkeley, Berkeley, CA). *ask1–2* yeast were a gift from Stephen Elledge's laboratory (Harvard University, Cambridge, MA). Yeast expressing *MTW1-mCherry* were a kind gift from Trisha Davis' laboratory (University of Washington, Seattle, WA). Plasmid pSB3218 (URA-CEN-sgRNA-Cas9) was a gift from Elçin Ünal's laboratory (University of California, Berkeley, CA).

### Culture conditions

Yeast were grown in yeast extract peptone dextrose (YPD) medium supplemented with 2% glucose (Sigma-Aldrich) and 0.02% adenine (MP Biomedicals) unless otherwise noted. For experiments involving *pGAL-MPS1*, cells were grown to mid-log ($OD_{600}$ =~1.2) in medium containing 2% Raffinose (Goldbio) and then supplemented with 4% galactose (Sigma-Aldrich) for 2.5 hours prior to harvest. All cells were grown at 23 °C except in the case of temperature sensitive alleles (*mps1–1* and *cdc15–2*) which were shifted to 37 °C for 2.5 hours prior to harvest. All cell cycle arrests were verified on a Nikon Eclipse E200 microscope equipped with a 40X objective (Nikon). For microscopy, yeast were grown in modified SC media, supplemented with 2% glucose and 0.06% adenine.

### Serial dilution assay

The designated *S. cerevisiae* strains were grown in YPD medium overnight. Cell concentration was measured on a spectrophotometer (Bio-Rad) and cells were diluted to $OD_{600}$ = 1.0. Next, serial dilutions (1:5) were made in water in a 96-well plate and the wells were then spotted

onto YPD or YPD + benomyl (Sigma-Aldrich) plates. Plates were grown at the indicated temperatures for 1–3 days prior to imaging.

## Auxin-inducible degradation

For auxin-induced degradation of *cdc20-AID*, cells expressing a *CDC20* C-terminal fusion of an auxin-response gene (*IAA7*) in the presence of *TIR1* were treated with 500 uM IAA (indole-3-acetic acid dissolved in DMSO, Sigma-Aldrich) for 2.5 hours prior to harvest or imaging.

## Pds1 time course assays

Cell cycle progression was monitored by analyzing the levels of the anaphase inhibitor Pds1/securin [66]. Mid-log cells were arrested in G1 with 1 μg/mL of α-factor. Cells were washed 3x in YPD medium and released into the cell cycle. Timepoints were collected (1 mL of cells) every 20 minutes for 140 minutes and α-factor was added back to cultures as cells began to form buds to prevent entry into the subsequent cell cycle. Whole cell lysates were made via bead pulverization (Biospec Products) in SDS sample buffer and protein concentration was normalized utilizing the yeast culture $OD_{600}$ as a reference.

## Microscopy

1 ml aliquots of cells were fixed with 3.7% formaldehyde in 100 mM phosphate buffer (pH 6.4) for 5 min, washed with 100 mM phosphate buffer (pH 6.4), permeabilized, and stained with DAPI by resuspending in 1.2 M Sorbitol/1% Triton X-100/100 mM phosphate buffer (pH 7.5) containing 1 μg/ml DAPI (Molecular Probes) for 5 min. Cells were then resuspended in the same buffer lacking DAPI. Cells were mounted on 1.5% agarose pads on slides and sealed with liquid VALAP (33% petroleum jelly, 33% lanolin, 33% paraffin) prior to imaging.

Cells were imaged on a Deltavision Ultra deconvolution high-resolution microscope equipped with a 100×/1.4 PlanApo N oil-immersion objective (Olympus) with a 16-bit scientific complementary metal–oxide–semiconductor detector at 22 °C. Cells were imaged in Z-stacks through the entire cell using 0.2 μM steps. Images were deconvolved in softWorX v7.2.1 (GE) using standard settings. Projections were made using a maximum-intensity algorithm. Composite images were assembled, and false coloring was applied with Adobe Photoshop.

## Fiji Analysis

For kinetochore fluorescence intensity quantification, using maximum intensity projections, sister kinetochores were circled in Fiji using the freehand tool and signal intensity was averaged between the kinetochore pair after dividing by the signal intensity of the fiducial kinetochore marker. Significance was assessed with a paired *t* test. For spindle fluorescence intensity quantification, spindles were circled using the freehand tool on maximum intensity projections of cells expressing GFP-Tub1. The ratio of Dad1-mKate2:Tub1-GFP, with each individual data point representing a single spindle, is shown in Fig 6D. Significance was assessed with a paired *t* test.

## Chromosome segregation and budding index assays

Cells growing in YPD medium were arrested in G1 with 1 μg/ml α-factor for 2.5 hours. Cells were washed twice in YPD, and released into medium lacking pheromone at 33 °C. For the budding index assay, 1mL aliquots of cells were taken every 20 minutes for 140 minutes. These cells were fixed and cellular buds were tabulated on a Nikon Eclipse E200 microscope equipped with a 40X objective (Nikon. α-factor was re-added after 60 minutes to prevent entry

into the subsequent cell cycle. For chromosome segregation analysis, cells with 256 *LacO* integrated near *CEN8* expressing a LacI-GFP fusion protein were grown, arrested, and released as above except that medium was supplemented with 1mM $CuSO_4$ as previously described [72]. Cells were fixed and imaged after 120 minutes and mis-segregation of Chr8 was quantified.

## Kinetochore protein purifications

Purifications were performed essentially as previously described [52,90]. Cells were grown to late-log ($OD_{600}$ =~3.0), harvested, resuspended in Buffer H (25mM Hepes, pH 8.0, 2mM $MgCl_2$, 0.1mM EDTA, 0.5mM EGTA, 0.1% NP-40, 15% glycerol, and 150mM KCl) containing phosphatase inhibitors (0.1mM Na-orthovanadate, 0.2μM microcystin, 2mM β-glycerophosphate, 1mM Na pyrophosphate, and 5mM NaF) and protease inhibitors (20μg/ml leupeptin, 20μg/ml pepstatin A, 20μg/ml chymostatin, and 200μM PMSF) and then flash frozen. Cells were lysed in a freezer mill (SPEX SamplePrep) and then ultracentrifuged at 98,500g for 90 minutes at 4°C. Lysate protein concentration was measured using Pierce BCA Protein Assay Kit (Thermo Scientific) and equivalent amounts of input protein were utilized for each experiment. Protein G Dynabeads (Invitrogen) were crosslinked with an α-M3DK antibody that recognizes the 3Flag epitope tag [91] (Genscript) and immunoprecipitations were performed at 4°C for 3 hours. Beads were washed once with lysis buffer containing 2mM DTT and protease inhibitors, three times with lysis buffer with protease inhibitors, and once in lysis buffer without inhibitors, and proteins were eluted by gentle agitation of beads in elution buffer (Buffer H + 0.5mg/ml 3FLAG Peptide [Sigma-Aldrich]) for 30min at room temperature. For Figs 1C (Dad1-Flag), 5A and 5B (Spc24-His-Flag), 250mM KCl was used in Buffer H throughout the purifications.

## Immunoblots and silver stains

Cell lysates were either made by beat-beat pulverization (Biospec Products) with glass beads in SDS buffer [24] or as described above for kinetochore protein purifications. SDS-PAGE gels were transferred to 0.2μm nitrocellulose membrane using a wet transfer apparatus (Hoefer). The following commercial antibodies were used for immunoblotting: α-PGK1 (Invitrogen; 4592560; 1:10,000), α-Flag (M2; Sigma-Aldrich; 1:3,000), and α-Myc (7D10; Cell Signaling; 1:1,000). The α-Ask1 antibody (1:3,000) was affinity purified from polyclonal Dam1c antibody serum that was previously described [57]. Briefly, GST-Ask1 was purified from *E. coli* using agarose glutathione beads (Sigma-Aldrich) and then coupled to Affigel 10 resin (Bio-Rad). α-Ask1 antibodies were then purified from the Dam1c serum using the Affigel column. The resulting α-Ask1 antibody was verified with immunoblot by examining the molecular weight difference between Ask1 and Ask1–13Myc in budding yeast lysates. The α-Ndc80-T248 phospho-antibody (1:1,000) was generated by Genscript using the peptide INQN{pTHR}QEITILSQPC. Verification of the antibody specificity is shown in Fig 2C. The α-Spc24 antibody (1:10,000) was generated by Genscript and verified by assessing a molecular weight shift from Spc24 to Spc24-His-3FLAG in yeast whole cell lysates. The phospho-antibodies α-Ndc80-pT54 and α-Ndc80-pT74 were generated and used as previously described [28]. The α-Ndc80 antibody (OD4), used at 1:10,000, was a kind gift from Arshad Desai (Department of Cellular & Molecular Medicine, University of California, San Diego, La Jolla, CA). Secondary antibodies used were sheep anti-mouse antibody conjugated to HRP (GE Biosciences) at 1:10,000 or donkey anti-rabbit antibody conjugated to HRP (GE Biosciences) at 1:10,000. Antibodies were detected using SuperSignal West Dura Chemiluminescent Substrate (Thermo Fisher Scientific). Immunoblots were imaged with a ChemiDock MP system (Bio-Rad). For silver stain analysis, protein samples were separated using precast 4–12% Bis-Tris gels

(Thermo Fisher Scientific) and stained using the Silver Quest Staining Kit (Invitrogen). For S1C Fig, proteins were separated on a 12% precast Bis-Tris gel (Thermo Fisher Scientific).

## Kinase and phosphatase assays

For kinase assays, kinetochores bound to beads (see kinetochore protein purifications) were washed into kinase buffer (50 mM Tris-HCl, pH 7.5, 75 mM NaCl, 5% glycerol, 10 mM $MgCl_2$, 1 mM DTT) with or without 10 μM ATP, incubated at 30°C for 15 min, eluted by boiling in SDS sample buffer and then analyzed via immunoblot. For phosphatase assays, Ndc80 complex was purified via Ndc80-Flag (see kinetochore protein purifications) and kept bound to Dynabeads (Invitrogen). Beads were washed once with phosphatase buffer (Buffer H, 1 mM $MnCl_2$), and then resuspended in phosphatase buffer with 200 U λ protein phosphatase (New England Biolabs) at 30 °C for 15 min. Control samples were treated identically minus the addition of phosphatase. Ndc80c was eluted by boiling beads in sample buffer containing SDS, and phosphorylation was analyzed via immunoblotting with α-Ndc80-T248 phospho-antibody (described above).

## Mass spectrometry

For purifications, see kinetochore protein purifications above. After the final wash step, beads were rinsed twice in pre-elution rinse buffer (50 mM Tris, pH 8.3, 75 mM KCl, and 1 mM EGTA) and eluted in 100 μl elution buffer (0.2% Rapigest [Waters Corp.] and 50 mM ammonium bicarbonate) with gentle agitation for 30 min at 20 °C. 10 μl of the sample was boiled in sample buffer and analyzed via silver stains. The remaining 90 μl was prepared for mass spectrometry by reduction with DTT (10 mM in 100 mM ammonium bicarbonate) at 56 °C for 45 min. The reduced protein solutions were alkylated with 2-chloroacetamide (55 mM in 100 mM ammonium bicarbonate) and incubated in the dark at ambient temperature for 30 min. 250 ng Trypsin (Promega) was added to the solutions and incubated overnight at 37 °C with mixing. Samples were acidified with 30% formic acid and mixed at room temperature for 1 h. Samples were spun down in a microfuge, and the supernatants were collected. All samples were desalted using ZipTip $C_{18}$ (Millipore) and eluted with 70% acetonitrile/0.1% trifluoroacetic acid. The desalted material was concentrated in a speed vacuum. The generated peptide samples were analyzed with a Thermo Fisher Scientific Easy-nLC 1000 coupled in line with a Thermo Fisher Scientific Orbitrap Fusion mass spectrometer. Mass spectrometry data were analyzed using Thermo Fisher Scientific Proteome Discoverer v2.4, with Sequest HT as the protein database search algorithm. The data were searched against an SGD yeast (UP000000589 from Oct. 25, 2019) database that included common contaminants (cRAPome; Jan. 29, 2015). Searches were performed with settings for the proteolytic enzyme trypsin, and maximum missed cleavages was set to 2. The precursor ion tolerance was set to 10 ppm, and the fragment ion tolerance was set to 0.6 D. Variable modifications included oxidation on methionine (+15.995 D) and phosphorylation on serine, threonine, and tyrosine (+79.966 D). Static modifications included carbamidomethyl on cysteine (+57.021 D). Peptide validation was performed using Percolator, and peptide identifications were filtered to a 1% false discovery rate.

For Figs 4C, S1E and S2D, the raw PSM counts for each kinetochore protein are presented divided by the raw PSM count for Dsn1 or Ndc80 (the immunoprecipitated protein) to produce "relative PSM" counts for each protein.

## Mps1 consensus motif analysis

Datasets from each sample were exported as Excel (.xlsx) files from Thermo Fisher Scientific Proteome Discoverer v2.4. We generated Python scripts to read and parse the.xlsx files which subsequently extracted the detected phosphorylation sites that contained a confidence score

$\geq 75\%$ and $> 1$ peptide-spectrum match (PSM) for each dataset. To assign a putative mitotic kinase responsible for each event, we used regular expressions (Python *re* and *regex* modules [https://pypi.org/project/regex/]) to match known kinase consensus motifs to each phosphorylation site. For this, we included Ipl1, Mps1, Cdc5, and Cdk1. The data from each dataset was compiled into a single table to easily compare across samples the presence of a phosphorylation event as well as a putative kinase (S2 Table). Additionally, we used regular expressions to search for all potential phosphorylation sites in all kinetochore proteins that match the aforementioned kinase consensus motifs regardless of their phosphorylation state and whether each site has been previously reported using the SGD. We note that not all phosphorylation reported is on the Saccharomyces Genome Database (S2 Table).

## Supporting information

**S1 Fig. Purifications to identify kinetochore phosphorylation via mass spectrometry.** (A) Silver stain of purifications of Dsn1-His-Flag kinetochore particles from WT (SBY8253) and *pGAL-MPS1* (SBY8810) genetic backgrounds. (B) Silver stain of purifications of Ndc80-Flag from WT (SBY10651) and *pGAL-MPS1* (SBY19721) genetic backgrounds. (C) Silver stain of purifications of Dad1-Flag from WT (SBY12441) and *pGAL-MPS1* (SBY20761) genetic backgrounds. * denotes proteins with molecular weight changes after Mps1 overexpression. (D) Silver-stained PAGE of purifications of Dsn1-His-Flag kinetochore particles from WT (SBY8253) and *mps1–1* (SBY8726) genetic backgrounds, both grown at 37 °C for 2.5 hours prior to harvest. (E) Mass spectrometry analysis of *pGAL-MPS1* purifications. The top 13 enriched proteins in the *pGAL-MPS1* background are shown as relative PSMs (i.e., raw PSMs divided by PSMs of the purified protein, either Dsn1 or Ndc80). Gene names denoted with a star (*) are related to actin biology.
(EPS)

**S2 Fig. Verification of Ndc80 phosphomutant protein levels and purification of kinetochore particles from Ndc80 phosphomutants.** (A) Lysates from WT (SBY3), $NDC80^{WT}$ (SBY21633), $ndc80^{T248A,T252A}$ (SBY21725), and $ndc80^{T248D,T252D}$ (SBY21768) yeast were analyzed via immunoblot with Ndc80 and Pgk1 (loading control) antibodies. (B) Silver-stained PAGE of purifications of Dsn1-His-Flag kinetochore particles from asynchronous WT (SBY21353), $ndc80^{T248A,T252A}$ (SBY21287), and $ndc80^{T248D,T252D}$ (SBY21289) genetic backgrounds. * denotes a heatshock background band. (C) Silver-stained PAGE of purifications of Dsn1-His-Flag kinetochore particles from anaphase arrested cells (*cdc15–2*) from $NDC80^{WT}$ (SBY22062), $ndc80^{T248A,T252A}$ (SBY22063), and $ndc80^{T248D,T252D}$ (SBY22011) genetic backgrounds. Cells were shifted to 37 °C for 2.5 hours prior to harvest. * denotes a heatshock background band. (D) Mass spectrometry of kinetochores from the genetic backgrounds in (C). Relative peptide spectrum match (PSM) counts (relative to Dsn1) are shown for kinetochore proteins.
(EPS)

**S3 Fig. Ndc80c localization in *ndc80* phosphomutant strains.** (A) Nuf2-mGFP signal was analyzed on both metaphase and anaphase kinetochores from asynchronous cultures of $NDC80^{WT}$ (SBY24287), $ndc80^{T248A,T252A}$ (SBY24288), and $ndc80^{T248D,T252D}$ (SBY24289) cells expressing Mtw1-mKate as a fiducial marker. Quantification of kinetochore signal is shown as the ratio of Nuf2-mGFP:Mtw1-mKate signal across 90 kinetochore clusters (3 biological replicates). Each point represents a kinetochore pair. Error bars represent SEM centered around the mean. n.s. indicates not significant compared to $NDC80^{WT}$. Metaphase: $ndc80^{T248A,T252A}$ p-value >0.54; $ndc80^{T248D,T252D}$ p-value > 0.97. Anaphase: $ndc80^{T248A,T252A}$ p-value >0.32; $ndc80^{T248D,T252D}$ p-value > 0.22. Metaphase mean values: $NDC80^{WT}$= 9.57±0.47, $ndc80^{T248A,T252A}$

= 10.08±0.60, $ndc80^{T248D,T252D}$ = 9.56±0.28; anaphase mean values: $NDC80^{WT}$= 9.49±0.06, $ndc80^{T248A,T252A}$ = 9.89±0.35, $ndc80^{T248D,T252D}$ = 8.85±0.63. (B) A representative image of a metaphase cell from each genotype analyzed in (A) is shown. Scale bar represents 2 μm. (C) A representative image of an anaphase cell from each genotype analyzed in (A) is shown. Scale bar represents 2 μm.
(EPS)

**S4 Fig. Dam1c and Ndc10 signal intensity in *ndc80* phosphomutant strains.** (A) Dad1-GFP signal intensity values from kinetochore clusters analyzed in Fig 5D. Lines represent mean values. Mean values: $NDC80^{WT}$ = 15397, $ndc80^{T248A,T252A}$ = 16748, $ndc80^{T248D,T252D}$ = 14678. (B) Ndc10-mCherry signal intensity values from kinetochore pairs analyzed in Fig 5D. Lines represent mean values. Mean values: $NDC80^{WT}$ = 1980, $ndc80^{T248A,T252A}$ = 2033, $ndc80^{T248D,T252D}$ = 1865.
(EPS)

**S5 Fig. Dam1 localization during the cell cycle and on anaphase spindles in *ndc80* phosphomutant strains.** (A) WT cells (SBY23116) expressing Dad1-mKate and GFP-Tub1 were released from a G1 arrest (alpha-factor) and a fraction of the cells were fixed and imaged every 20 minutes. A representative image of the population is shown for each timepoint as the cells progress through mitosis. Scale bar represents 2 μm. (B) Representative image of a late anaphase WT (SBY23417) cell expressing Dad1-GFP and Mtw1-mKate from an asynchronous culture. Scale bar represents 2 μm. (C) GFP-Tub1 signal intensity on anaphase spindles from Fig 6D. Mean values: $NDC80^{WT}$ = 5182, $ndc80^{T248A,T252A}$ = 5773, $ndc80^{T248D,T252D}$ = 4070. * indicates p-value <0.001 compared to $NDC80^{WT}$ (D) Dad1-mKate signal intensity on anaphase spindles from Fig 6D. Mean values: $NDC80$ = 401, $ndc80^{T248A,T252A}$ = 558, $ndc80^{T248D,T252D}$ = 359. (D) Spindle lengths (μm) from anaphase cells used for quantification in Fig 6D. n.s. indicates not significant compared to $NDC80^{WT}$ ($ndc80^{T248A,T252A}$ p-value >0.45; $ndc80^{T248D,T252D}$ p-value>0.82) Mean values: $NDC80^{WT}$ =12.2 μm, $ndc80^{T248A,T252A}$ = 12.5 μm, $ndc80^{T248D,T252D}$ = 12.3 μm.
(EPS)

**S6 Fig. Characterization of anaphase spindles in *ask1–2 ndc80* phosphomutants.** (A) Serial dilutions of WT (SBY3), *dad1–1* (SBY22736), $ndc80^{T248A,T252A}$ *dad1–1* (SBY22738), and $ndc80^{T248D,T252D}$ *dad1–1* (SBY22740) were plated on YPD and grown at the specified temperatures. (B) WT (SBY22771), *ask1–2* (SBY22852), $ndc80^{T248A,T252A}$ *ask1–2* (SBY22853), and $ndc80^{T248D,T252D}$ *ask1–2* (SBY22854) cells, all containing *GFP-TUB1* and *MTW1-mKate2* were released from G1 after alpha-factor treatment and anaphase spindle phenotypes were quantified as cells progressed through mitosis. Each individual experiment (n>200 anaphase spindles) is represented as a symbol above the bar graph. Error bars represent SEM. * indicates p-value <0.0001, n.s. indicates not significant (p-value >0.16), both compared to $NDC80^{WT}$ *ask1–2*. Mean values: WT = 1.4%±0.1, $NDC80^{WT}$ *ask1–2* = 6.8%±1.4, $ndc80^{T248A,T252A}$ *ask1–2* = 16.8%±0.6, $ndc80^{T248D,T252D}$ *ask1–2* = 7.3%±2.7. (C) Representative images of WT and mutant ("curved/uneven") anaphase spindles from (B) are shown. Scale bar represents 2 μm.
(EPS)

**S1 Table. Phosphorylation of budding yeast kinetochore proteins detected by mass spectrometry with putative kinases denoted.**
(DOCX)

**S2 Table. Output table generated via the supplementary code which contains the analysis of kinetochore phosphorylation from all mass spectrometry experiments presented in Figs 1 and S1.** "Detected phos" tab is the code output containing analysis of all mass spectrometry data

presented in Figs 1 and S1. "Kinase stats" tab contains statistics on the number of detected phosphorylation events mapping to each consensus motif in addition to scraping of SGD for previously unreported phosphorylation events. "All pred. kin. sites" tab lists all putative kinase consensus motifs found in each yeast kinetochore protein based on the regular expressions referenced above. "Regex used" tab contains the regular expressions utilized to identify consensus motifs for mitotic kinases in budding yeast.
(DOCX)

**S3 Table. Yeast strains used in this study.**
(DOCX)

**S4 Table. Plasmids used in this study.**
(DOCX)

**S5 Table. Oligonucleotides used in this study.**
(DOCX)

## Acknowledgments

We are grateful to the Biggins' lab for critical reading of the manuscript and to the members of the Asbury laboratory and Seattle Mitosis Group for fruitful discussion. We thank the Proteomics & Metabolics shared resource facility at Fred Hutchinson Cancer Center for mass spectrometry sample processing and data analysis. We thank Georjana Barnes, Trisha Davis, Arshad Desai, Stephen Elledge, Kevin Hardwick, Adele Marston, David Morgan, and Elçin Ünal for providing reagents.

## Author contributions

**Conceptualization:** Christian R Nelson, Darren R Mallett, Sue Biggins.

**Data curation:** Christian R Nelson, Darren R Mallett, Sue Biggins.

**Formal analysis:** Christian R Nelson, Darren R Mallett, Sue Biggins.

**Funding acquisition:** Sue Biggins.

**Investigation:** Christian R Nelson, Darren R Mallett.

**Methodology:** Christian R Nelson, Darren R Mallett, Sue Biggins.

**Project administration:** Sue Biggins.

**Resources:** Christian R Nelson, Darren R Mallett, Sue Biggins.

**Software:** Darren R Mallett.

**Supervision:** Sue Biggins.

**Validation:** Christian R Nelson, Darren R Mallett, Sue Biggins.

**Visualization:** Christian R Nelson, Darren R Mallett, Sue Biggins.

**Writing – original draft:** Christian R Nelson, Darren R Mallett, Sue Biggins.

**Writing – review & editing:** Christian R Nelson, Darren R Mallett, Sue Biggins.

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
