## [Decision Letter · Decision Letter 0]

21 Nov 2024

PGENETICS-D-24-01251Spindle integrity is regulated by a phospho-dependent interaction 

between the Ndc80 and Dam1 kinetochore complexesPLOS Genetics Dear Dr. Biggins, Thank you for submitting your manuscript to PLOS Genetics. Three experts reviewed the manuscript and found that it makes a useful contribution to the field of chromosome segregation. However, they have raised important questions about some of the experiments and the overall model envisioned by the authors. They also suggest specific experiments to clarify and expand the conclusions, and raise issues regarding the genetic background in which the effects of Ndc80 mutants are observed. They also suggest better quantitation of signals obtained using antibodies. Therefore, we invite you to submit a revised version of the manuscript that addresses the points raised during the review process. Please submit your revised manuscript within 60 days Jan 20 2025 11:59PM. If you will need more time than this to complete your revisions, please reply to this message or contact the journal office at plosgenetics@plos.org. Please include the following items when submitting your revised manuscript:

* A rebuttal letter that responds to each point raised by the editor and reviewer(s). You should upload this letter as a separate file labeled 'Response to Reviewers '. This file does not need to include responses to any formatting updates and technical items listed in the 'Journal Requirements' section below. * A marked-up copy of your manuscript that highlights changes made to the original version. You should upload this as a separate file labeled 'Revised Manuscript with Track Changes '. * An unmarked version of your revised paper without tracked changes. You should upload this as a separate file labeled 'Manuscript '. If you would like to make changes to your financial disclosure, competing interests statement, or data availability statement, please make these updates within the submission form at the time of resubmission. Guidelines for resubmitting your figure files are available below the reviewer comments at the end of this letter. We look forward to receiving your revised manuscript. Kind regards, Ashok Bhagwat, Ph.D.Academic EditorPLOS Genetics Pablo WappnerSection EditorPLOS Genetics

Aimée Dudley

Editor-in-Chief

PLOS Genetics

Anne Goriely

Editor-in-Chief

PLOS Genetics

**Journal Requirements:**

3) We notice that your supplementary Figures, and Tables are included in the manuscript file. Please remove them and upload them with the file type 'Supporting Information'. Please ensure that each Supporting Information file has a legend listed in the manuscript after the references list.

4) Please amend the description of the main file to be "manuscript" rather than "cover letter."

5) Please note that your Data Availability Statement is currently missing the DOI/accession number of each dataset OR a direct link to access each dataset. If your manuscript is accepted for publication, you will be asked to provide these details on a very short timeline. We therefore suggest that you provide this information now, though we will not hold up the peer review process if you are unable.

1) State what role the funders took in the study. If the funders had no role in your study, please state: "The funders had no role in study design, data collection and analysis, decision to publish, or preparation of the manuscript.".

7)  Please ensure that the funders and grant numbers match between the Financial Disclosure field and the Funding Information tab in your submission form. Note that the funders must be provided in the same order in both places as well. Currently, this funder "Howard Hughes Medical Institute" is missing from the Financial Disclosure.

Please indicate by return email the full and correct funding information for your study and confirm the order in which funding contributions should appear. Please be sure to indicate whether the funders played any role in the study design, data collection and analysis, decision to publish, or preparation of the manuscript.

**Reviewers' comments:**Reviewer's Responses to Questions

**Comments to the Authors:**

Reviewer #1: In their paper titled “Spindle integrity is regulated by a phospho-dependent interaction between the Ndc80 and Dam1 kinetochore complexes”, authors Nelson, Mallett, and Biggins examine the yeast kinetochore phosphoproteome and mechanistically dissect the function of two previously uncharacterized phosphosites on the kinetochore protein Ndc80. They first used mass spectrometry on purified kinetochores and identified many previously unidentified phosphorylation sites. Many of these sites fit the consensus sequences for the main mitotic kinases: Ipl1, Mps1, and Cdc28. Mps1 mutants cannot accurately biorient chromosomes in metaphase. However, none of the kinase’s substrate’s phosphosite mutants have fully recapitulated this phenotype, leading the authors to explore the novel Mps1 targets in the hopes of identifying biorientation mutants. They focus on two phosphosites in the MT attachment protein Ndc80. The authors first show that these sites are phosphorylated by Mps1 in vitro and in vivo. They then characterize the phenotype of non-phosphorylatable and phosphmimic mutations for the two sites. They find that there is not effect on chromosome segregation. However, there is some change to Dam1 complex localization and the non-phosphorylatable mutants are synthetic lethal with hypomorphic mutations in the Dam1 complex. This combination of mutants leads to some defects in spindle morphology in metaphase and anaphase. They conclude that a soluble pool of Ndc80 regulates Dam1 function at the mitotic spindle.

Overall, the thorough identification of phosphosites in the yeast kinetochore provides a valuable resource to the community. While some of the phenotypes that they find for the two Ndc80 sites are intriguing, the evidence for their model of soluble pools of Ndc80 regulating Dam1 complex localization is not fully founded.

Major issues:

1. The authors find that sensitization with hypomorphic Dam1c mutants leads to growth defects and/or lethality in the T248A,T252A background. This is a clear result that could potentially provide a strong clue to the role of the phosphorylation. However, it is not clear to me how a slight increase in the percentage of spindles with morphological defects would lead to such a phenotype. Do these morphologic changes somehow lead to cell arrest and/or death? It would be worth revisiting potential kinetochore functions in this background. Is there frequent chromosome missegregation with these combinations of mutants? Does the checkpoint get activated?

2. On a related note, does the Ndc80 T248A,T252A mutant alone affect spindle morphology or is this only observed in the context of the adk1-2 allele? This information would be useful in determining if the spindle morphology phenotypes correlate with the growth defects. Such morphological changes could also impact the intensity measurements for Dad1 on the spindle (Figure 6D).

3. Spindle localization intensity for MAPs can vary greatly depending on spindle length in anaphase. For figure 6D, spindle length should be measured to show that it does not change between the measured spindles for the different mutants or it should be made clear that only anaphase spindles within a narrow range of lengths were used for quantification.

4. I do not know of any evidence for the soluble forms of Ndc80c and Dam1c interacting. To propose this as a model, they should show this interaction either with purified complexes or with pull-downs from yeast without kinetochores.

5. In figure 3B, the degree of saturation for the different antibodies in the western blots is quite variable. The increase for pT74 and pT54 phosphorylation is subtle, so the oversaturation with the pT248 antibody may be obfuscating a similar increase. A lower exposure time for the pT248 antibody should be shown.

Minor issues:

6. Just because none of the identified Mps1 phosphorylation sites has yet to recapitulate the biorientation issue, that does not necessarily mean that the correct sites have not been found. It could simply mean that there is a lot of redundancy between them, and one would need to mutate most of them in the same strain to get the biorientation phenotype. The authors could consider mentioning this as an alternative theory if they agree.

7. The benomyl dilution series in Figure 4A is missing a positive control.

Reviewer #2: This manuscript describes a series of experiments analysing phosphor-regulation of kinetochore complexes, in particular Mps1-dependent regulation of Ndc80 and Dam1C. Novel phenotypes are observed showing that phosphorylation of Ndc80 on T248 and T252 control its interaction with Dam1C. Interestingly these point to a function of Dam1c in organising and stabilising the anaphase spindle. The data is of excellent quality and is very nicely and logically presented. I am happy to recommend this manuscript for publication once minor changes have been made.

Lines 126-7. It is worth pointing out that these cells will be metaphase-arrested.

Lines 139-140 (SFig 1E). Upon Mps1 overexpression, lots of MS hits with the actin cytoskeleton were observed. More discussion of this would be interesting.

Line 201. Were Abs to T252 also attempted to be generated? Can the authors tell if both Thr residues are modified on a single polypeptide, or is it usually one or the other? What ratio?

Fig 3B. Are there ways to specifically induce error-correction activity (rather than use nocodazole) in Fig3B?

Lines 272-4 At times residue T252 is somewhat ignored in the authors’ description/discussion.

Fig 5C. Please add representative images of the A and D mutants

Fig 6B. Perhaps a 3rd repeat of this experiment would generate significant numbers for the alanine mutant too?

SFig 4. Why is the GFP-tubulin intensity reduced in D mutant? More discussion needed on this. Has it been observed in related, relevant mutants?

Line 338 should read Supplementary Fig 4B-D.

Line 353. Can an alternative phrase be used, replacing ‘nearly dead’?

Line 427-9. More speculation on bundling/oligomerisation functions of Dam1C would be interesting.

Line 443. Has Dam1C been purified from anaphase cells to look for spindle interactors?

Reviewer #3: Nelson et al address the question how phosphorylation at kinetochores ensures faithful chromosome segregation. They conducted mass-spectrometry experiments on purified kinetochore particles to identify phosphorylated residues at the kinetochore, particularly after overexpression of the Mps1 kinase. They focus on the Ndc80 residues T248 and T252, phosphorylated by Mps1 in metaphase. They characterize phospho-null ndc80 T248A T252A and phosphomimetic T248D T252D mutants and find that the alleles do not impair growth, mitotic progression, or kinetochore composition in a wildtype background. They identify ndc80 T248D T252D to show a higher affinity for the Dam1 complex in anaphase, which correlates with decreased Dam1 content on anaphase spindles. In combination with ts alleles of the Dam1 complex subunits Dad1 and Ask1, Ndc80 T248A T252A shows growth defects and alterations in spindle architecture both in metaphase and anaphase.

Although the characterized ndc80 alleles do not have a strong phenotype, this work represents a meaningful addition to open questions concerning Mps1’s contributions to mitotic fidelity. Thus, the work will be of interest to the field. The experiments are conducted on a very high technical level, logically presented and thoughtfully discussed.

I have a couple of points that should be addressed:

Main points:

While the mapping of the phospho-sites and the demonstration of mps1-dependence is very convincing, the model in the end remains somewhat unclear. The question is, where does the phospho-regulation of the Ndc80-Dam1c interaction occur? The phosphorylation sites are derived from kinetochore purifications, so it would be logical to assume that Mps1 phosphorylates kinetochore-bound Ndc80 to promote Dam1 association. The authors state, however, that the characterized Ndc80 alleles regulate a kinetochore-independent Dam1c pool. Do they think the responsible pool of Ndc80c is kinetochore bound?. If yes: how do they imagine the kinetochore bound Ndc80c regulate Dam1c at another localization? If not, where is the Ndc80 pool localized?

One obvious thing to do is to localize the ndc80 mutant alleles themselves in fluorescence microscopy and analyze kinetochore and spindle association. Maybe ndc80 T248D T252D with increased Dam1 binding could lead to co localization of the complexes on anaphase spindles and this may somehow cause a problem? The authors should test this.

The authors find reduced spindle labelling with GFP-Tub1 in ndc80 T248D T252D mutants. Does this result in spindle phenotypes, e.g. broken spindles? This should be analyzed or stated if not. It would be nice to show respective micrographs. Do the authors think the interaction between Dam1c and Ndc80c regulates tubulin incorporation/dynamics at spindles? The authors should discuss this.

Minor point:

Figure 2B: I’d show a slightly larger portion of Ndc80/Nuf2, to put the T248 and T252 sites into a better structural context. Alphafold models should do a good job of predicting this.

**Have all data underlying the figures and results presented in the manuscript been provided?**

Reviewer #1: Yes

Reviewer #2: Yes

Reviewer #3: Yes

PLOS authors have the option to publish the peer review history of their article (what does this mean? ). If published, this will include your full peer review and any attached files.

**Do you want your identity to be public for this peer review?** For information about this choice, including consent withdrawal, please see our Privacy Policy .

Reviewer #1: No

Reviewer #2: No

Reviewer #3: No

---

## [Decision Letter · Decision Letter 1]

6 Mar 2025

Dear Dr Biggins,

We are pleased to inform you that your manuscript entitled "Spindle integrity is regulated by a phospho-dependent interaction 

between the Ndc80 and Dam1 kinetochore complexes" has been editorially accepted for publication in PLOS Genetics. Congratulations!

However, one of the reviewer did have some helpful suggestions to improve your manuscript further and I urge you to consider them.

Yours sincerely,

Ashok Bhagwat, Ph.D.

Academic Editor

PLOS Genetics

Pablo Wappner

Section Editor

PLOS Genetics

Aimée Dudley

Editor-in-Chief

PLOS Genetics

Anne Goriely

Editor-in-Chief

PLOS Genetics

Comments from the reviewers (if applicable):

Reviewer's Responses to Questions

**Comments to the Authors:**

Reviewer #1: The authors addressed all of my concerns with convincing new experiments and informative edits to the text.

Reviewer #2: The authors have done an excellent job, responding to the reviewers suggestions and criticisms.

This is a high quality piece of work and I am happy to recommend publication.

Reviewer #3: Spindle integrity is regulated by a phospho-dependent interaction

between the Ndc80 and Dam1 kinetochore complexes" (PGENETICS-D-24-01251R1)

The authors have addressed my remaining questions with a careful revision. I am fully supportive of publication.

1. They provide evidence that the interaction can be requlated in the absence of kinetochores, using a kinetochore-null ndc10-1 allele. (Figure 5E).

2. The conducted additional microscopy experiment, testing the localization of Ndc80c to anaphase spindles. (Supplementary Figure 3).

3. they provide better explanations for the hypomorphic nature of GFP-Tub1 effects.

I would still suggest to optimize display of the location of the mutant residues in Figure 2. Below a quick suggestion using pdb 5TCS (Ndc80c dwarf) The superposition with the human structure (2ve7) could be omitted as the position of Hec1 S202 is not quite the same.

**Have all data underlying the figures and results presented in the manuscript been provided?**

Reviewer #1: Yes

Reviewer #2: Yes

Reviewer #3: None

PLOS authors have the option to publish the peer review history of their article (what does this mean? ). If published, this will include your full peer review and any attached files.

**Do you want your identity to be public for this peer review?** For information about this choice, including consent withdrawal, please see our Privacy Policy .

Reviewer #1: No

Reviewer #2: No

Reviewer #3: No

**Data Deposition**

http://datadryad.org/submit?journalID=pgenetics&manu=PGENETICS-D-24-01251R1

**Press Queries**

---

## [Editor Report · Acceptance letter]

PGENETICS-D-24-01251R1

Spindle integrity is regulated by a phospho-dependent interaction 

between the Ndc80 and Dam1 kinetochore complexes

Dear Dr Biggins,

We are pleased to inform you that your manuscript entitled "Spindle integrity is regulated by a phospho-dependent interaction 

between the Ndc80 and Dam1 kinetochore complexes" has been formally accepted for publication in PLOS Genetics! Your manuscript is now with our production department and you will be notified of the publication date in due course.

With kind regards,

Anita Estes

PLOS Genetics

On behalf of:
